# Enhancing generalizability of data-driven urban flood models by incorporating contextual information

Tabea Cache[1], Milton Salvador Gomez[1,2], Tom Beucler[1,2], Jovan Blagojevic[3], João Paulo Leitao[4], and Nadav Peleg[1,2]

[1]Institute of Earth Surface Dynamics, University of Lausanne, Lausanne, Switzerland
[2]Expertise Center for Climate Extremes, University of Lausanne, Lausanne, Switzerland
[3]Institute of Environmental Engineering, ETH Zurich, Zurich, Switzerland
[4]Department of Urban Water Management, Swiss Federal Institute of Aquatic Science and Technology, Dubendorf, Switzerland

**Correspondence:** Tabea Cache (tabea.cache@unil.ch)

**Abstract.** Fast urban pluvial flood models are necessary for a range of applications, such as near real-time flood nowcasting or processing large rainfall ensembles for uncertainty analysis. Data-driven models can help overcome the long computational time of traditional flood simulation models, and the state-of-the-art models have shown promising accuracy. Yet the lack of generalizability of urban pluvial flood data-driven models to both unseen rainfall and distinctively different terrain, at the fine resolution required for urban flood mapping, still limits their application. These models usually adopt a patch-based framework to overcome multiple bottlenecks, such as data availability and computational and memory constraints. However, this approach does not incorporate contextual information of the terrain surrounding the small image patch (typically 256 m x 256 m). We propose a new deep-learning model that maintains the high-resolution information of the local patch and incorporates a larger surrounding area to increase the visual field of the model with the aim of enhancing the generalizability of urban pluvial flood data-driven models. We trained and tested the model in the city of Zurich (Switzerland), at a spatial resolution of 1 m, for 1-hour rainfall events at 5 min temporal resolution. We demonstrate that our model can faithfully represent flood depths for a wide range of rainfall events, with peak rainfall intensities ranging from 42.5 mm h$^{-1}$ to 161.4 mm h$^{-1}$. Then, we assessed the model's terrain generalizability in distinct urban settings, namely Luzern (Switzerland) and Singapore. The model accurately identifies locations of water accumulation, which constitutes an improvement compared to other deep-learning models. Using transfer learning, the model was successfully retrained in the new cities, requiring only a single rainfall event to adapt the model to new terrains while preserving adaptability across diverse rainfall conditions. Our results indicate that by incorporating contextual terrain information into the local patches, our proposed model effectively simulates high-resolution urban pluvial flood maps, demonstrating applicability across varied terrains and rainfall events.

## 1 Introduction

Urban pluvial flooding represents a global threat to population and infrastructure that is expected to increase as floods become more frequent and the world's population grows, with 68% of the world population projected to live in cities by 2030 (UN,

2018). From an economic point of view, the concentration of wealth in urban areas combined with accelerated infrastructure development has led to a great increase in economic losses from floods (Kundzewicz et al., 2014) and these losses are projected to further increase globally (Winsemius et al., 2016).

Along with the growing exposure of population and assets, the occurrence of pluvial floods is projected to increase due to both climate change and the effects of urbanization (IPCC, 2022). Pluvial flooding occurs in response to intense precipitations that cause the failure of the drainage system. Due to global warming, short-duration extreme rainfall, which is the key trigger to pluvial flooding, is intensifying globally (Fowler et al., 2021; Tabari et al., 2020), and this intensification can be exacerbated by the urban environment (Han and Baik, 2008; Huang et al., 2022; Li et al., 2020; Liang and Ding, 2017). It is also well
established that urbanization increases the occurrence of pluvial flooding by modifying the hydrological response: sealed impervious surfaces and reduced vegetation decrease infiltration capacity, surface storage, and evapotranspiration, resulting in higher peak discharges and runoff volumes (Leopold, 1968; Miller et al., 2014; Semadeni-Davies et al., 2008; Kaspersen et al., 2017).

 The foreseen urbanization and climate change, and its projected impacts on urban pluvial floods, encourage the development
of resilient cities (Ahmed et al., 2018; Berndtsson et al., 2019; Rosenzweig et al., 2018). While there exists a consensus regarding the increase in urban pluvial flood risk (Houston et al., 2011; Kundzewicz and Pińskwar, 2022), the extent of increased risk and its attributed causes still constitute major knowledge gaps (IPCC, 2022; Kundzewicz et al., 2014). Loss databases are not suited for risk trend analysis due to biases from improvements in reporting, changes in vulnerability, and the inability to distinguish amongst the factors (climatic or non-climatic) triggering the hazard (Peduzzi et al., 2012; Willems et al., 2012).
Thus, there is a need to model the changes in extreme rainfall due to changes in climate and urban areas, and the impact they will have on the flood regime in each city individually.

 Extreme short-duration rainfall events can be modeled using physically-based climate models with high spatial resolution (such as convection-permitting models; Dallan et al., 2023), stochastic-mechanistic climate models (Peleg et al., 2017), or stochastic-statistical methods (Marra et al., 2019, 2024). The outputs of these models can be used as inputs to numerical
hydrodynamic models, which are the most robust and reliable models for estimating urban hydrological responses to rainfall (Kourtis and Tsihrintzis, 2021). However, these models are also characterized by a long computational time. While recent improvements in computational power and more efficient algorithms have reduced the burden of hydrodynamic models, their run times are still insufficient for applications requiring a high number of simulations. This is problematic, as multiple runs of these models are required per city to account for the large degree of uncertainty in future climate projections and urban
development scenarios (Hirsch, 2011; Miller and Hutchins, 2017), necessitating the development of alternative models.

 The use of machine learning for fast flood mapping has been given growing attention in recent years (Nearing et al., 2021). Models based on convolutional layers have demonstrated the potential to emulate urban pluvial flood maps as they can best extract spatial information characterizing the flood events (Bentivoglio et al., 2022). To increase the amount of training data, and address memory limitations of handling large images, these models operate on local patches rather than the entire catchment
area (Berkhahn and Neuweiler, 2024; do Lago et al., 2023; Guo et al., 2021, 2022; Löwe et al., 2021; Seleem et al., 2023). The patch-based model presented by Guo et al. (2021), for example, can predict water depths 1,400 times faster than traditional

hydrodynamic models, as demonstrated for a range of rainfall events in the cities of Zurich, Luzern, and Coimbra. Löwe et al. (2021) developed a model for urban pluvial flood mapping and evaluated its prediction performance in a city in Denmark at 5 m resolution, reserving approximately 25% of the area for validation and testing. Although these areas were included in the validation, they were excluded from training, and the model still performed well in these areas. In another study, Guo et al. (2022) assessed the terrain generalizability of a data-driven flood model across 656 catchments in Switzerland. The model was able to adapt to new catchments, yet it did not incorporate rainfall as an input, limiting its predictions to the single rainfall event used for generating the training flood maps. More recent advances include the work of Seleem et al. (2023), in which a convolutional neural network (CNN) based approach was compared with a random forest approach for urban pluvial flood mapping in three study areas in Berlin at 1 m spatial resolution. The authors found that the CNN model could benefit from transfer learning to enhance performance in terrain on which the model was not trained. Generalizability to terrain was also investigated by do Lago et al. (2023), who developed a conditional generative adversarial network that distributes a previously known runoff volume over a given catchment. The generator was able to identify cells where the water level exceeds 0.3 m, and to predict the water levels for cells below that threshold. Berkhahn and Neuweiler (2024) have used autoencoders to compress data contained in flood maps and a recursive time series prediction model to simulate water depth time series in urban areas, at 6 m resolutions. However, they did not consider generalizability to terrain. Lastly, another promising application of machine learning for rapid flood mapping is the use of hybrid approaches, which combine the advantages of different model types. The advantages of hybrid approaches have been demonstrated in recent studies, including Fraehr et al. (2023), where a fast model was developed by combining a simplified, physics-based hydrodynamic model, optimized for speed through a coarse computational grid and long computational time steps, with a mathematical model that transforms the flood patterns from the low-fidelity model into those of high-fidelity, non-simplified models. The model's generalizability was tested in two study areas with distinct topographies, for a temporal resolution of 1 h and a spatial resolution of 20 m.

Despite the advancements made in recent years in developing data-driven models for flood predictions, there are still some major challenges to overcome. One of them is the generalizability of the models to unseen case studies, including both unseen rainfall and distinctively different terrain, at the fine resolution required for urban flood mapping. This limits their application to real-case studies.

Although the patch-based approach that the high-resolution urban pluvial flood data-driven models adopt overcomes multiple bottlenecks (e.g., amount of training data and memory limitations of handling large images), it ignores contextual information from the surrounding terrain that can be crucial for flood mapping (Fig. 1). In order to preserve global elevation information, Guo et al. (2022) have investigated the resizing-based option that down-samples the input and then up-samples the outputs to the original size. This option can process larger areas, yet it causes significant information loss which makes it a less optimal method for urban flood mapping that requires high spatial resolution. Including larger context information while preserving the high-resolution local representation of the patch is a common issue in the field of image segmentation (e.g. biomedical and land-use/land-cover image segmentation; Alsubaie et al., 2018; BenTaieb et al., 2017; Li et al., 2021; Mou et al., 2020; Shaban et al., 2019). Combining multi-scale information in context-aware models has been shown to improve image segmentation performance together with keeping models computationally efficient (Sirinukunwattana et al., 2018).

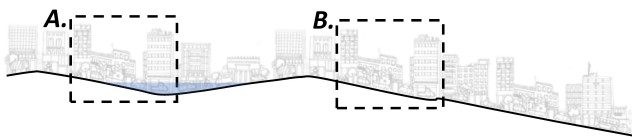

**Figure 1.** Schematic showing the importance of contextual information of local patches to predict urban pluvial flood maps; local terrain does not contain the information necessary to distinguish the flood responses in patches A and B.

Here, we present the development of a new context-aware data-driven model for high-resolution urban pluvial flood mapping and investigate its performance, generalizability, and transfer learning abilities, using the cities of Zurich, Luzern, and Singapore as case studies.

## 2 Context-aware flood model description

We developed a context-aware data-driven model for urban pluvial flooding. The objective of the model is to extract and combine the information from the high-resolution local patch, its surrounding terrain (or context) and the rainfall times series to emulate the corresponding flood map. To achieve this, we developed a joint model that couples different types of neural networks and learns dependencies between the local patch and its surrounding area. The model consists of the following components (Fig. 2, Fig. S1): (i) three convolutional encoders that extract latent information from the multi-scale spatial features (i.e. high-resolution local patch and lower resolution contextual information); (ii) an attention mechanism that measures the correlation between the local patch and its context; (iii) a recurrent neural network (RNN) that analyzes the hyetograph; and (iv) a decoder that converts the extracted information from both the terrain and rainfall data into the flood depth prediction. The various components of the model are explicitly defined hereinafter.

### 2.1 Multi-scale terrain features

To help inform the model of spatial features that govern water accumulation, terrain information images are derived from the digital elevation model (DEM) for the local patch and its context at different scales. To reduce computational costs, context images are rescaled to the same size as the local patch (256 x 256) using the Lanczos downsampling filters. Assuming a native resolution of the DEM of 1 m, the visual field of the model thus covers surfaces of 512 m x 512 m and 1024 m x 1024 m at spatial resolutions of 2 m and 4 m respectively (see the 'Inputs' and 'Spatial features' panels in Fig. 2).

### 2.2 Multi-scale convolutional encoders

The multi-scale image patches, consisting of the terrain features derived from the DEM, are fed to convolutional encoders that are composed of stacked convolutional layers and pooling layers (Fig. 2, Fig. S1). These operations reduce the spatial dimensionality of the input images while extracting latent information contained in an increasing number of feature maps.

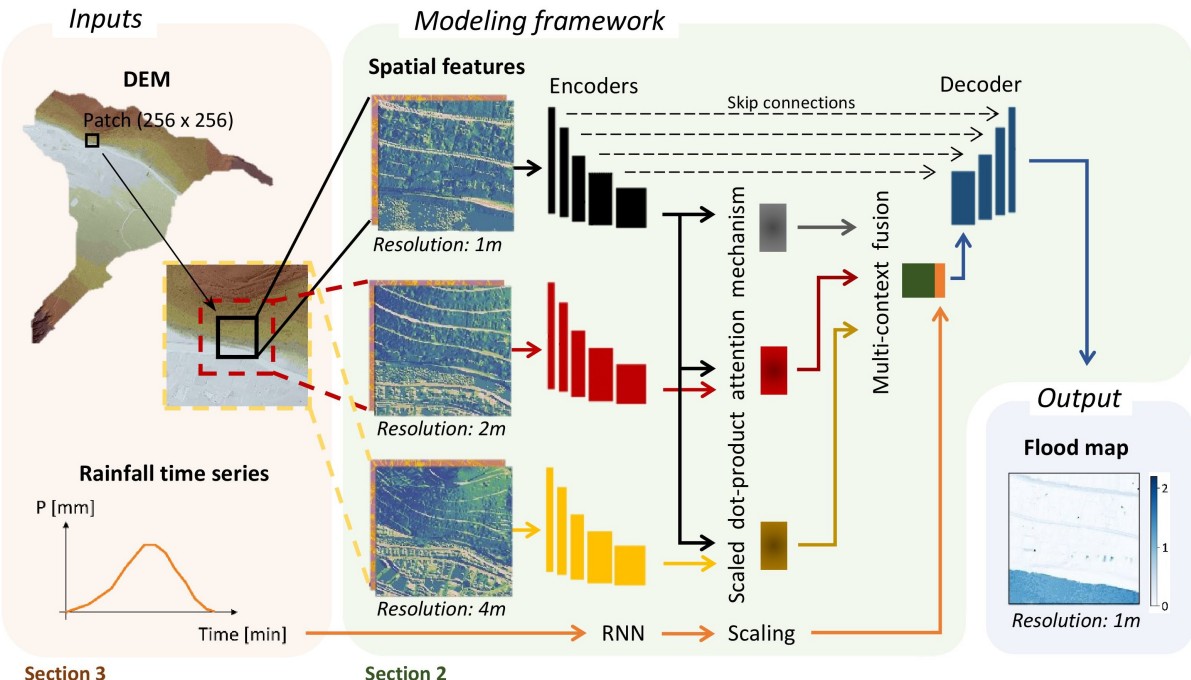

**Figure 2.** Schematic diagram showing the inputs, modeling framework, and output of the data-driven flood model. A detailed diagram of the model architecture is presented in Fig. S1.

Our model includes three distinctive encoder networks with the same architecture, each processing multi-channel images at different resolutions.

### 2.3    Contextual attention mechanism

An attention mechanism then associates the features of the local patch with the features of the context to capture the relationships between the high- and lower-resolution images (see 'Scaled dot-product attention mechanism' in Fig. 2). This atten-
tion mechanism is similar to the locality-aware contextual correlation module applied by Li et al. (2021) for high-resolution geospatial image segmentation (Fig. S1). By applying the scaled dot-product attention to the encoder extracted features, the significance of each multi-channel image features are weighted to help the model correctly combine contextual information.

### 2.4    Hyetograph processing

Additionally, information from the dynamic input are extracted using a base RNN layer. This is a type of neural network that is
known to be well suited for handling time series data, especially for the task of rainfall-runoff modeling (Géron, 2019; Kratzert et al., 2019). RNNs can process time series of arbitrary length, which allows to use the model for rainfall events of different durations. This enables to address the limit of use of the model to rainfall events of one specific duration, as is the case in

similar studies (e.g., Guo et al., 2021). The output of the RNN is then scaled by the normalized accumulated rainfall. This multiplicative scaling ensures that the rainfall forcing from the RNN is proportional to the accumulated rainfall, resulting in zero forcing when there is zero rainfall. The scaling is defined for rainfall event $i$ as:

$S_i = \frac{P_{i,acc}}{P_{norm}}$, with $P_{i,acc} = \sum_t P_{i,t}$ and $P_{norm} = \sum_t P_{min,t}$ where t is the time step and $P_{min}$ refers to the rainfall event with minimum accumulated rainfall in the training set. We normalized the scaling in order to avoid vanishing or exploding gradients issues. Ultimately, this procedure led to a model with better performance (Table S2).

## 2.5 Fused terrain and hyetograph features upsampling

Lastly, the scaled RNN output is concatenated with the locality-aware features extracted from the scaled dot-product attention. The resulting combination serves as the input to the decoder (Fig. 2). Similarly to the encoder, the decoder consists of stacked upsampling layers, each comprising one deconvolution (convolutional transpose) layer followed by two convolution layers (Fig. S1). Each upsampling layer of the decoder is joined with the corresponding features of the local patch from the encoder through a skip connection, in a similar way to the UNet model (Ronneberger et al., 2015). The latter model has recently gained attention in hydrological studies (Guo et al., 2022; Löwe et al., 2021; Seleem et al., 2023) as it is particularly efficient at localizing, and thus at processing images in which spatial information is important. The operations of the decoder progressively upsample the fused feature maps into the flood map in a way that exploits complementary information from the local patch, the different contexts, and the rainfall time series.

## 2.6 Hyperparameters

Following initial tests, we adopted the Mean Squared Error (MSE) as the loss function to train the model:

$MSE = \frac{1}{n}\sum_{i=1}^{n}(y_i - \hat{y}_i)^2$, where $y_i$ represents the true values, $\hat{y}_i$ represents the predicted values, and $n$ is the number of observations. We implemented an early stopping callback to regularize the model, i.e. to avoid overfitting. This callback terminates the model's training when the performance on the validation set is not improving for a certain number of epochs, defined by the patience. We applied the Adam optimizer (Kingma and Ba, 2014) with a learning rate of 0.0001 and implemented the Glorot normal initialization strategy (Glorot and Bengio, 2010), which is a way to avoid unstable gradients when training the model. The kernel sizes of the convolutional layers were 3 x 3 and 2 x 2 for the pooling and deconvolutional layers, the activation function of all layers was Leaky-ReLU following Guo et al. (2021), and the batch size was 32.

## 2.7 Urban flood map predictions

After completing the training process, the model can be used for making predictions. The flood maps of the entire city are constructed by assembling the local patch predictions. The terrain patches are extracted at regular grid intervals ensuring that the whole city is covered. To produce more robust predictions, the patches are extracted at a grid size distance of half the patch size so that the flood map patches overlap. In these overlapping areas, the final prediction is the average of the patch predictions

in that area. This method was found to be the best option by Guo et al. (2021) as it gives a good balance between accuracy and prediction time.

## 3 Model input specifications

### 3.1 Data specifications for model training

The model requires two types of inputs: static inputs (multi-channel terrain images) and dynamic inputs (rainfall hyetographs). Furthermore, the desired output, i.e., the target flood map, must also be included in the training data. Hereafter, we describe the pre-processing framework for the training data assuming a native resolution of the DEM of 1 m.

First, the DEM is upscaled to 2 m and 4 m, and spatial features are derived from the DEM at the different spatial resolutions (1 m, 2 m, and 4 m) before being stacked as multi-channel images. The spatial features used to train the neural network were chosen based on previous studies (Guo et al., 2021; Löwe et al., 2021) and the model's performance in initial tests. We found (not shown) that feeding the model with the DEM, mean curvature, aspect (sine and cosine), depth of the sinks and the slope (in radians) helped the model learn best, suggesting that these features can encapsulate the hydrological characteristics of the catchment related to the dynamics of water during floods. Additionally, these features can all be derived directly from the DEM, thus eliminating the need for further data, such as imperviousness maps for example. We extract the spatial features from the DEM using the RichDEM library (Barnes, 2016).

Patches are then extracted from the multi-channel images at random locations, with the constraints of a maximum overlap threshold of 20% between two patches and a minimum study area coverage of 10% for each individual patch. The corresponding output patches are extracted from the target flood map. The pairs of input and output patches are subsequently augmented. The model should be equivariant to rotation and flip transformations, i.e., these transformations of the input and output patches are arbitrary, as long as they are consistently applied to input and output pairs. Hence, since the augmentation techniques need to be applied to both inputs and outputs, and that rainfall-runoff follows non-linear relationships, we can apply 7 augmentation techniques consisting of a combination of flips and 90° rotations of the images (Fig. S2). This enables to increase the amount of training data while limiting the patch overlap and thus limiting the risk of overfitting the model. In fact, similar studies with comparable study area sizes have used disproportionate amounts of patch locations without data augmentation, thus extracting redundant patches and facing the risk of large overlaps between training and validation patches (Fig. S3, Guo et al., 2021; Seleem et al., 2023). Hence, these models could be overfitting to the terrain, even though this would not be apparent when comparing the training and validation losses using standard evaluation metrics (Section S1, Table S1).

Lastly, all multi-channel image patches are transformed through min-max scaling. This transformation consists of forcing the input to have the same scale, here [-1, 1], by shifting and rescaling the data. It is commonly applied to machine learning data as machine learning algorithms do not perform well on inputs with very different scales (Géron, 2019). We found that the model performed best when the normalization was applied on each patch individually. We also tested its application across the entire study area, i.e., extracting patches after normalizing the feature images of the full study area, similar to previous studies (Guo et al., 2021; Löwe et al., 2021). While this could help preserve some information about the position of the patch in its

larger context, it also forces patches to have values falling in a very small range (e.g., full study area DEM with values between 0 and 1, and DEM patches with values between 0.455 and 0.495), therefore considerably decreasing the performance of the model (Table S2).

Before being fed to the model, some data must be reserved for validation and testing purposes. To facilitate this, both patch locations and rainfall time series are partitioned into training, validation, and testing datasets. First, the rainfall events must be partitioned in a way that ensures independence among training, validation, and testing rainfall events, with respective proportions of 67%, 11% and 22%. Then, the patch characteristics (i.e., patch location and patch augmentation combinations) are randomly divided into training (90%) and validation (10%) sets. Lastly, some of the training data, consisting of the combination of both patch characteristics and rainfall events, are allocated to the validation dataset. Following this workflow, the data in the training and validation sets are allocated in an 80%-20% ratio. This partitioning strategy ensures that the testing rainfall events remain unseen by the model until evaluation, thereby maintaining the validity of the rainfall generalizability assessment. Consequently, the validation set includes the following combinations of data: (1) new rain and new terrain patch, (2) training rain and new terrain patch, and (3) new rain and training terrain patch.

To summarize, the model's inputs are three multi-channel image patches (one 256 x 256 x 6 image for the local patch at 1 m resolution and two 256 x 256 x 6 images for the context at 2 m and 4 m resolutions), along with the unprocessed rainfall time series and the corresponding target flood map patch (256 x 256 image covering the same area as the local patch at 1 m resolution).

## 3.2    Data specifications for flood map generation

The data requirement and pre-processing framework for using the presented model to simulate flood maps closely resemble those employed during training, with some minor deviations. The main differences lie in the fact that; the target output is not informed to the model, the locations of the patches are neither sampled randomly nor split into train/test/validation sets, and no augmentation technique is applied to the patches. The model's inputs are three multi-channel image patches of the terrain (each with dimensions 256 x 256 x 6 and resolutions 1 m, 2 m, and 4 m, assuming a native DEM resolution of 1 m) and the rainfall hyetograph for which the user wants to produce the flood map. The patches are sampled at a regular grid interval of half the patch size (here 128), ensuring a comprehensive coverage of the study area. The model will subsequently simulate flood maps for all patches and reconstruct the flood map for the entire study area by combining the output patches.

## 4    Model training and transfer learning

First, we trained and tested the model in the city of Zurich and evaluated its performance to represent flood depths for a wide range of rainfall events. Second, the model's terrain generalizability was assessed in distinct urban settings, namely Luzern and Singapore.

In an effort to enhance the model's performance in new cities, we assessed the suitability of employing transfer learning. Transfer learning is a popular approach to improve the training of deep computer vision models by using the knowledge of

**Table 1.** Training and testing overview.

| Training | | | Testing on new rainfall events | |
|---|---|---|---|---|
| Terrain | Rainfall | Model initialization | Terrain | Results presentation |
| Zurich | 12 events | None | Zurich | Fig. 4, Fig. 5 |
| | | | Luzern | Fig. 6b |
| | | | Singapore | Fig. 6b |
| Luzern | 1 event | Model trained for Zurich | Luzern | Fig. 6c, Fig. 7, Fig. 8 |
| Singapore | 1 event | Model trained for Zurich | Singapore | Fig. 6c, Fig. 7, Fig. 8 |

existing models that perform similar, or identical, tasks to the new model. By initializing or freezing some of the weights and biases of the layers of the new model with the ones from the existing pre-trained model, this technique speeds up the training

of the new model and requires significantly less training data to retrain the new model (Erhan et al., 2010; Géron, 2019) as there is no need to train the model from scratch.

## 5    Rainfall and terrain generalizability

### 5.1    Dataset and training details

The training, testing, and validation datasets for Zurich were extracted from 18 flood maps at 1 m resolution for a catchment

of 12.7 km$^2$ (Fig. 3 Guo, 2019). These flood maps were generated using the cellular automata model WCA2D (Guidolin et al., 2016, implemented in CADDIES-caflood) and correspond to 18 1-hour spatially uniform rainfall events at 5 min resolution with mean intensities ranging from 19 mm h$^{-1}$ to 46 mm h$^{-1}$. These correspond to rainfall events with return periods ranging from 2- to 100-y in Zurich. Each return period is associated with three events that have different hyetograph shapes and maximum intensities (Fig. 3). To ensure the model's equivariance to zero-padding and its ability to handle rainfall events of differing

durations, we randomly selected three rainfall events with varying mean intensities and introduced random zero-padding at the beginning and end of these events. Lastly, we introduced an event with zero rainfall and a corresponding flood map showing no flooding. The neural network can thus learn to distinguish the effects of rainfall events with different characteristics. The DEM used as input for these simulations did not include buildings' representation. We extracted a total of 1,250 patches for training and validation of the model (Fig. S3), and the rainfall events were split into train/test sets so that each return period

was represented only once in the test dataset, similar to Guo et al. (2021). Furthermore, the rainfall events in the test set are independent from the ones in the training set as they have different shapes and peak intensities.

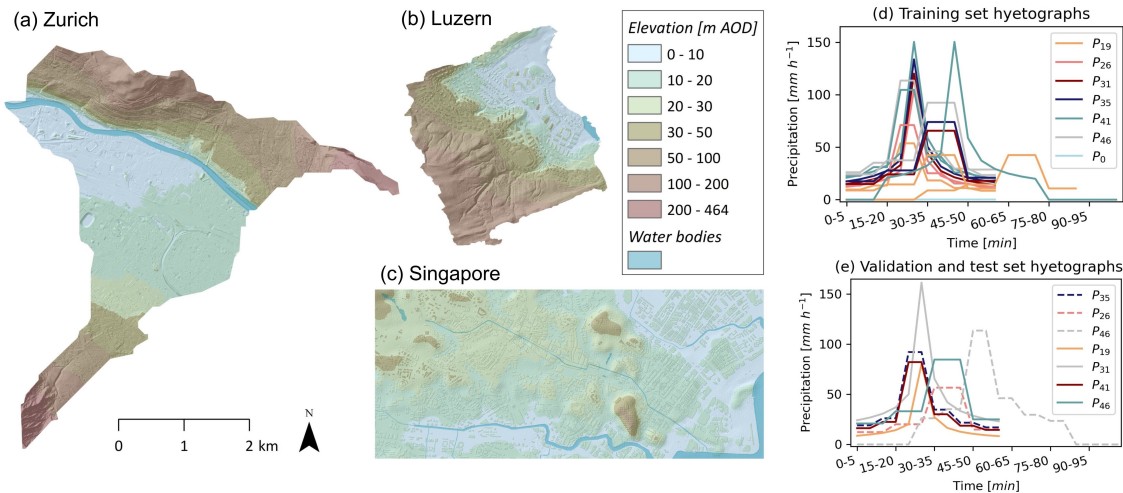

**Figure 3.** DEM of (a) Zurich, (b) Luzern, and (c) Singapore, and hyetographs used to (d) train, and (e) validate and test the model. The elevation datum of the DEM is set to the minimum elevation in each of the study areas respectively. The validation and test hyetographs are shown by the dashed and solid lines respectively.

## 5.2 Generalizability to rainfall events

Our model's performance was evaluated by comparing the emulated flood maps for the entire city with those simulated in CADDIES-caflood. Hereafter, we will refer to these as simulated and target flood maps respectively, and denote the rainfall events by their mean intensity $m$ and shape $s$ as follows: $P_{m-s}$. The shape notation ranges from 1 to 3, where 1 denotes the events with the highest peak rainfall intensity and 3 denotes the most distributed event types.

First, we evaluated the model's ability to accurately predict water depths in Zurich for rainfall events in the test set (Fig. 4). We chose to show the flood maps for the least and most intense rainfall events as these can reflect the performance of the model to distinguish situations where limited flooding occurs, as well as the prediction performance in extreme conditions.

The visual comparison of the target and predicted flood maps suggests that the model successfully reproduces the spatial pattern of water accumulation for both low- and high-intensity rainfall events. This is confirmed by the Root Mean Squared Error (RMSE, defined as $RMSE = \sqrt{MSE}$) of the predicted flood maps for $P_{19-1}$ and $P_{46-1}$, which are respectively 15.3·10⁻³ m and 19.6·10⁻³ m.

Furthermore, the model accurately identifies cells below 0.1 m (Fig. 5). The critical success index (CSI), which measures the accuracy of the predictions, is defined as the ratio of correctly identified cells (i.e. true positives, TP) to the sum of correctly identified cells, missed target cells (i.e. false negatives, FN), and incorrectly identified cells (i.e false positives, FP):

$$CSI = \frac{TP}{TP + FN + FP} \tag{1}$$

The CSI for water depths below 0.1 m, i.e. where positive values in Eq. 1 correspond to water depths below 0.1 m, are 0.98 for $P_{19-1}$ and 0.97 for $P_{46-1}$. Additionally, the majority of the cells in the target flood maps fall below the 0.1 m threshold, representing respectively 84.6% and 80.8% of all cells (Fig. 5). To address this imbalance and evaluate the prediction performance of the model above the critical 0.1 m wet threshold (Seleem et al., 2023; Kaspersen et al., 2017), we also evaluated the RMSE values for cells exceeding 0.1 m in the target flood maps ($RMSE_{0.1}$; Table S4). The wet cells $RMSE_{0.1}$ values for $P_{19-1}$ and $P_{46-1}$ are respectively $52.9 \cdot 10^{-3}$ m and $44.7 \cdot 10^{-3}$ m.

Fig. 4, which shows a zoomed-in area of size 650 x 650 greater than the aggregation size of 128 x 128, enables us to visually evaluate the smoothness of the predictions at the boundaries of the patches. The absence of artifacts such as horizontal and vertical lines confirms that the multi-scale patch-based predictions along with the patch aggregation method produce continuous flood maps. This suggests that merging contextual information with the local patch alleviates the issue of single-scale patch models cutting off and disconnecting hydrological objects such as flow paths or sinks.

The model's prediction performance was further analysed by investigating the relative error in a set of water depth ranges (Fig. 5) for the least and most intense rainfall events, as well as for an intermediate event, $P_{31-2}$, with mean intensity approximately equal to the average mean intensity from the least and most intense events. The median relative error is fairly even across water depth ranges and across rainfall events. For all rainfall events, we observe a trend towards underprediction of the water levels when moving to the largest target water depth ranges. The error is the lowest for the most intense rainfall event $P_{46-1}$, which exhibits the smallest median residual error, and the lowest inter-quartile range for water depths above 0.3 m.

We also compared the performance of the model relative to some terrain characteristics, similar to Guo et al. (2021). We likewise focused the analysis on the most extreme condition from the test set, i.e., the $P_{46-1}$ rainfall event. While the previous study found that their model performed worse in downstream areas (lowest 33% terrain elevations) than in upstream areas (highest 33% terrain elevations), we found that our multi-scale model improves performance in downstream areas, bringing the error in these areas to a similar range as the error in upstream areas (Table S3). Our model also exhibits a major prediction improvement in depressions, with the errors first and third quartile reduced by a factor of 3 compared to the state-of-the-art model (Table S3). This suggests that adding contextual information helps the model predict more accurately water routing and accumulation in lower laying areas or terrain depressions.

Overall, the results show that our model can faithfully reproduce flood depths for a wide range of unseen rainfall events for the terrain on which the model was trained, here Zurich.

## 5.3 Generalizability to terrain

Next, we verified that the model can predict flood maps in new, unseen terrain. We tested the model in two cities: Luzern (Switzerland) and Singapore (Fig. 3, Table 1). The former has a similar landscape type to Zurich. Singapore, on the other hand, is not located in a mountainous environment and therefore presents a much flatter topography than Zurich. In both cases, the DEM of the urban areas included the representation of the built environment while this was not the case for Zurich. The spatial resolution of the DEMs are 1 m for Luzern and 2 m for Singapore, meaning the multi-channel image patches for Singapore

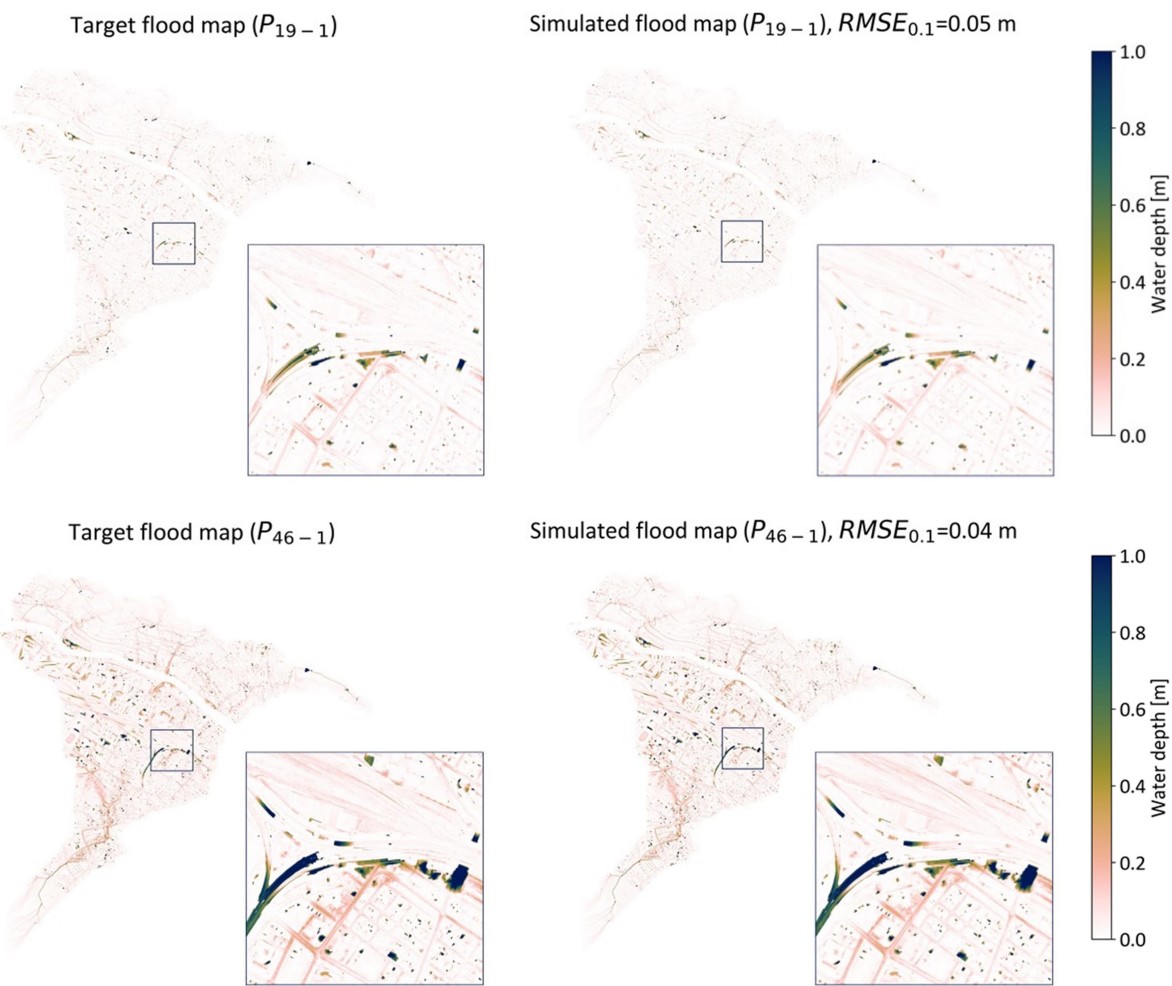

**Figure 4.** Target (left) and simulated (right) flood maps in Zurich for the least ($P_{19-1}$) and most ($P_{46-1}$) intense rainfall events in the test set.

have resolutions of 2 m, 4 m, and 8 m. This allows to also test the abilities of the model to adapt to terrain data at different resolutions.

We present the target and simulated flood maps in Luzern and Singapore for the most intense rainfall event in the dataset, i.e. $P_{46-1}$, in Fig. 6a. and 6b. We used the model presented heretofore to simulate the flood maps in 6b., i.e., the model trained for Zurich. Despite the differences in terrain, and especially the representation of the built environment in the DEMs and the spatial resolution, the model broadly captures the areas of water accumulation and the flood hazard levels in both new cities. This suggests that the contextual information, along with the consistent data pre-processing, helps the model extract information relevant to flood mapping in unseen terrain (Fig. S4). In fact, we identify continuous flooded areas that are larger

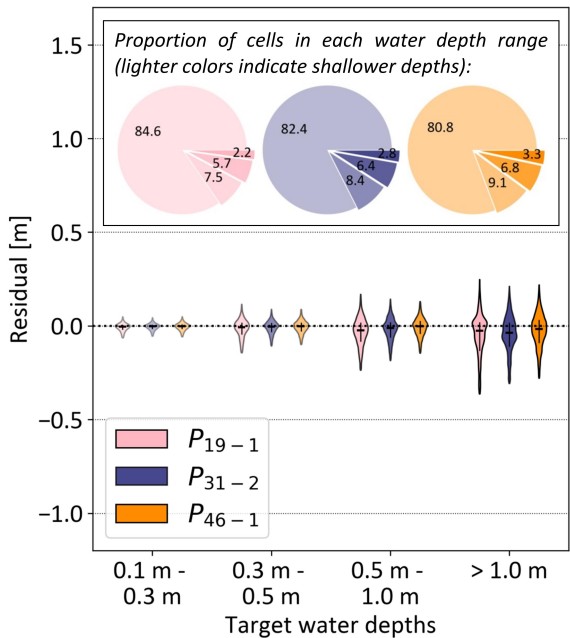

**Figure 5.** Violinplot of the simulation error in Zurich, expressed as the difference between the target flood map and simulated flood map, for different target water depth ranges and different rainfall events ($P_{19-1}$, $P_{31-2}$ and $P_{46-1}$, that have respectively lowest, intermediate, and highest mean intensities of the test set). The vertical black lines range from the 25th to 75th percentiles and the horizontal black line indicates the median value. Negative values correspond to the underprediction of the simulations. The pie charts illustrate the proportion of cells in each water depth range in the target flood maps. The water depth ranges are indicated by different shading levels: lower water depths are represented by more transparent colors, while higher water depths are depicted with darker colors.

than the local patch size. However, while the spatial patterns are broadly reproduced, the model fails at correctly predicting the water depth magnitudes.

To evaluate how well the model detects the locations of water accumulation, we evaluated the CSI for wet and flooded cells (Table S4). Considering a wet cell depth threshold of 0.1 m and a flood depth threshold of 0.3 m (i.e. positives values in Eq. 1 correspond to flood depths above 0.1 m and 0.3 m respectively), we obtained the following CSI values for $P_{46-1}$: $CSI_{0.1, \text{Luzern}}$ = 0.50, $CSI_{0.3, \text{Luzern}}$ = 0.32, $CSI_{0.1, \text{Sgp}}$ = 0.48, and $CSI_{0.3, \text{Sgp}}$ = 0.35. The $CSI_{0.3}$ are lower than the $CSI_{0.1}$ mainly because of a decrease in true positives, meaning the model detects fewer flooded cells than it detects wet cells. This is due to the model's water depth underprediction, resulting in fewer cells reaching the flooded depth threshold. Despite the more pronounced terrain differences between Zurich and Singapore and the different spatial resolutions of the terrain data, with Singapore having a 2 m DEM and both Zurich and Luzern having 1 m DEMs, the CSI values indicate that the model's performance is similar in Luzern and Singapore.

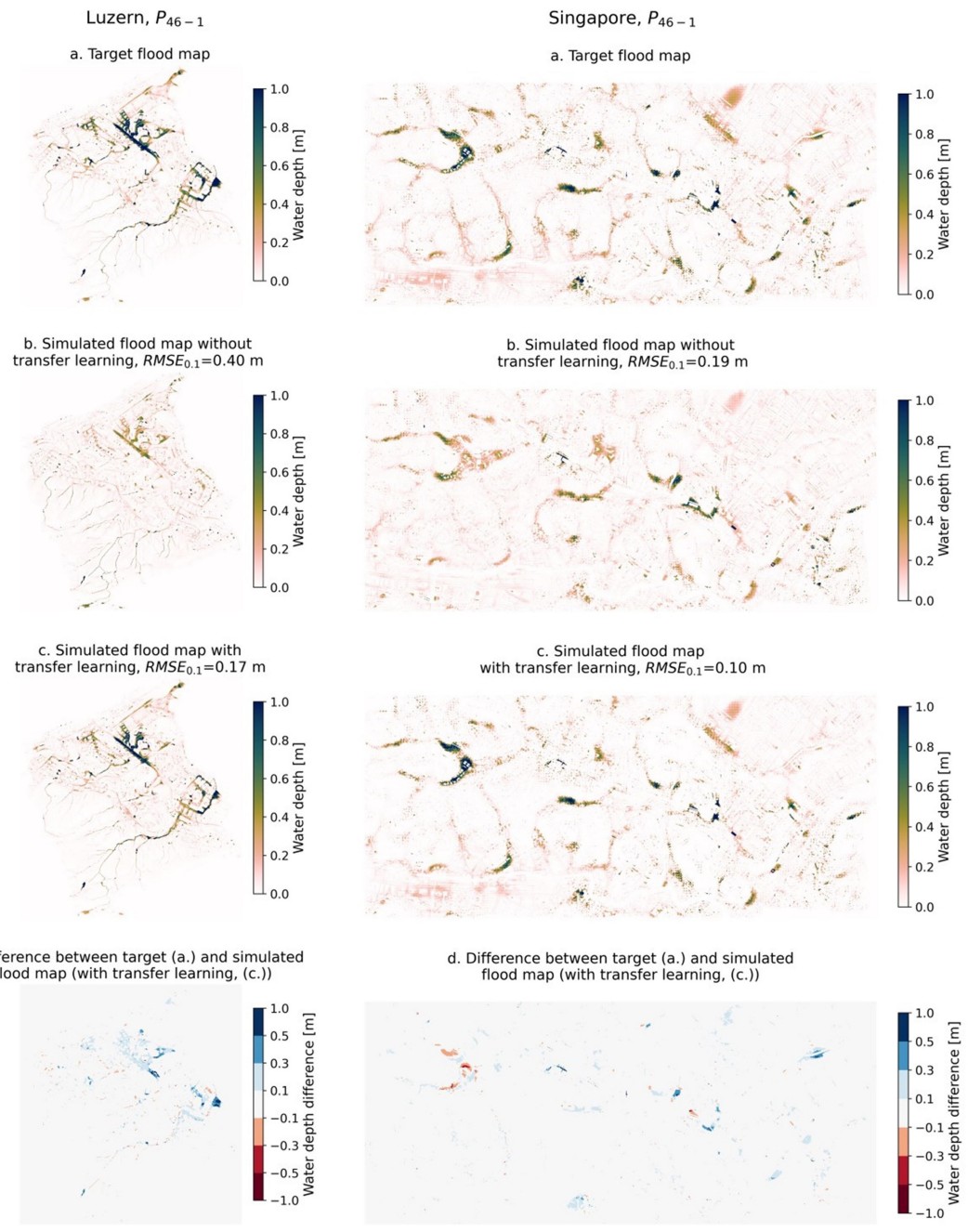

**Figure 6.** Flood maps and error in Luzern (left) and Singapore (right). Note that the models used to simulate the flood maps 6c were retrained for $P_{31-2}$.

## 6 Improving generalizability to terrain through transfer learning and parsimonious retraining

While the results obtained when applying the model to terrain in which the model has not been trained are promising, the errors in water depth magnitudes are too high to consider that the model generalizes well (see Fig. 6b). Therefore, we investigated the effectiveness of transfer learning to improve our multi-scale urban pluvial flood model performance.

Here, we considered the model trained for Zurich as the pre-trained model and transferred its knowledge to models that we separately trained for our case studies; Luzern and Singapore (Table 1). Note that Luzern and Singapore are case studies and that the presented framework could potentially be applied to any other city. As the objective of this study is to develop a model that is fast and limits as much as possible computations for the end-user, the additional training data necessary should be either readily available or fast to produce. Consequently, the models were retrained using only one rainfall event and its corresponding flood map for each respective city. The model is thus solely informed of the response of the new terrain to one rainfall event.

### 6.1 Case study 1: A resembling terrain

#### 6.1.1 Terrain details

We first evaluated how well the model can adapt to a terrain resembling the one for which the model was initially trained. To that end, we retrained the model in Luzern. Like Zurich, Luzern is located in Switzerland and has a mountainous landscape. The size of the study area is 6.3 km$^2$ and the spatial resolution of the DEM and target flood maps is 1 m. The target flood maps for Luzern were also generated in CADDIES-caflood. Considering the size of the study area in Luzern, we reduced the number of training patches to 620 (Fig. S3).

The model was trained on only one rainfall event and its corresponding flood map. The model's training hyperparameters were exactly the same as for the model trained in Zurich, except that all the model's layers were initialized with the layers of Zurich's model and the patience was set to 1. This enabled the prevention of overfitting for the specific rainfall event on which the model was retrained, considering that the aim is to adapt the model to the new terrain while preserving adaptability across diverse rainfall conditions.

#### 6.1.2 Model performance

First, we evaluated the performance of the model retrained in Luzern for event $P_{31-2}$. We selected this event to retrain the model as its mean intensity lies midway between the lowest and highest mean intensities of all the events. Additionally, the shape of the event is neither the sharpest one nor the most uniform one. This enabled the evaluation of the model's extrapolation ability towards less and more extreme events, both in terms of mean and maximum intensity of the events.

Comparing the simulated flood maps visually and through various performance metrics shows that the new model can accurately reproduce the flood maps in Luzern, and that the model consistently outperforms the predictions from the model trained on Zurich (Fig. 6c, Fig. 7a, Fig. S5, Table S4). Fig. 6c shows the flood map simulated for rainfall event $P_{46-1}$. The model

accurately reproduces the spatial distribution of water accumulation and the corresponding water depths. The simulated flood map achieves an $RMSE_{0.1}$ of 0.17 m and the CSI values for wet and flooded cells reach $CSI_{0.1, \text{Luzern}} = 0.72$ and $CSI_{0.3, \text{Luzern}} = 0.68$. Furthermore, Fig. 6d shows that the areas where the performance error of the model are highest are located at the boundary with a water body (Fig. 3) or in areas where the model had successfully predicted high water levels (Fig. 6c).

We further analysed the model's prediction performance for the events with lowest, intermediate, and highest mean intensities, respectively $P_{19-1}$, $P_{31-2}$, and $P_{46-1}$, according to different target water depth ranges (Fig. 7a). The violinplot shows that our model accurately reproduces the water depths for all water levels. The relative prediction error is the highest in the cells with the largest water depths for all rainfall events, and the model tends to underpredict the water levels. In fact, the median error in the cells with target flood depths higher than 1 m lies between -8 cm and -22 cm. However, considering the high water

depths in which these errors occur, the absolute median relative error does not exceed 15%. Furthermore, we can notice that the prediction error is lower for $P_{31-2}$ than for $P_{19-1}$ and $P_{46-1}$. This result is in line with the fact that the model was trained for the event $P_{31-2}$. Overall, the model can faithfully reproduce the flood maps in Luzern for unseen rainfall events.

     Second, we evaluated the model's extrapolation ability by comparing the $RMSE_{0.1}$ of the predicted flood maps according to the rainfall event used to retrain the model (Fig. 8a). We retrained the model for the following rainfall events: $P_{19-1}$, $P_{41-3}$ and

$P_{46-1}$. Subsequently, we simulated the flood map for each of these rainfall events and $P_{31-2}$ using all four retrained models. The prediction performances are summarized in the heatmap in Fig. 8a. The asterisk that indicates the lowest $RMSE_{0.1}$ for each prediction rainfall is located along the diagonal. This means that each model had the lowest $RMSE_{0.1}$ for the prediction rainfall event for which it was trained, and that the lowest $RMSE_{0.1}$ of each prediction rainfall was achieved by the model which was trained for this same rainfall event. In line with expectations, the $RMSE_{0.1}$ increases as the mean rainfall intensity

moves away from the training rainfall mean intensity. Furthermore, the results suggest that the model extrapolates better when the prediction rainfall event mean intensity is smaller than the training event mean intensity.

## 6.2    Case study 2: A distinct terrain

### 6.2.1    Training details

Next, we evaluated how well the model can adapt to a terrain that is distinct from the one for which the model was initially

trained. We chose to retrain the model in Singapore, which is an island city with one of the most high-density urban development in the world. The terrain is much flatter than in Zurich or Luzern, with a maximum elevation in the study area reaching 56 m (against a 464 m maximum change in elevation in Zurich). The size of the study area is 15.4 $km^2$ and the spatial resolution of the DEM and target flood maps is 2 m. The target flood maps for Singapore were also generated in CADDIES-caflood. We extracted 450 patches to keep a similar patch density as in Zurich and Luzern. The model was trained on only one rainfall event

and its corresponding flood map after initializing the model's layers with the layers from the model trained for Zurich. The hyperparameters were kept unchanged, except for the patience that was set to 1 to prevent overfitting to the rainfall event for which the model was retrained.

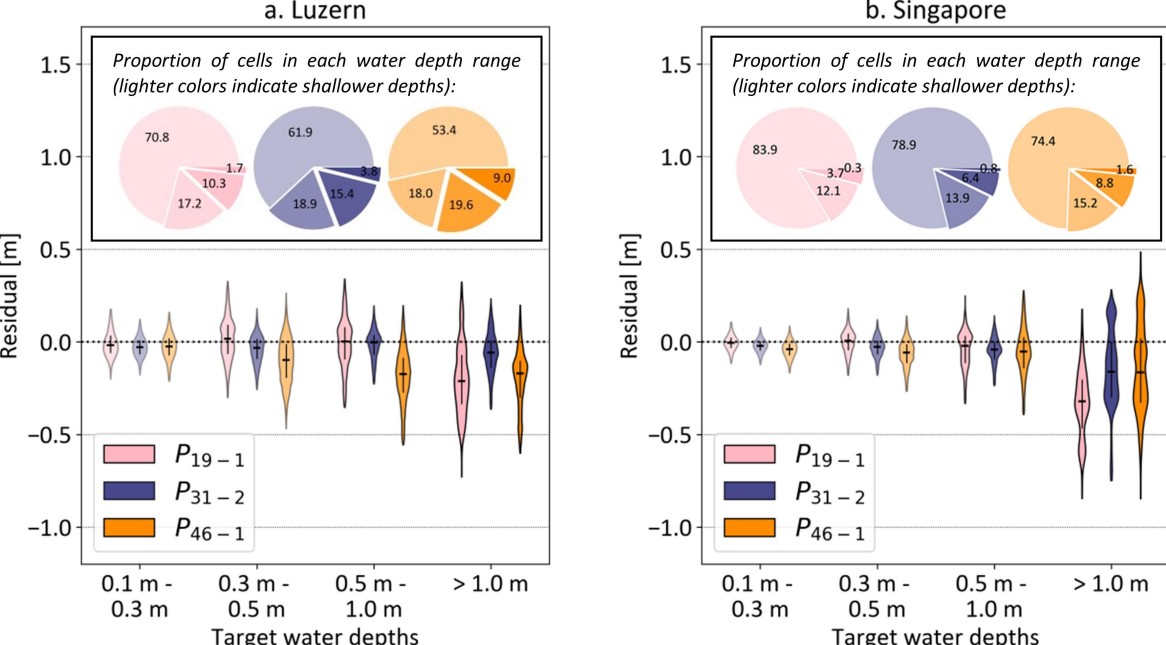

**Figure 7.** Violinplot of the simulation error in (a.) Luzern and (b.) Singapore, using the models retrained for the event $P_{31-2}$ in each city respectively. The error is expressed as the difference between the target flood map and simulated flood map, for different target water depth ranges and different rainfall events. The vertical black lines range from the 25th to 75th percentiles and the horizontal black line indicates the median value. Negative values correspond to the underprediction of the simulations. The pie charts illustrate the proportion of cells in each water depth range in the target flood maps. The water depth ranges are indicated by different shading levels: lower water depths are represented by more transparent colors, while higher water depths are depicted with darker colors.

### 6.2.2 Model performance

We evaluated the performance of the model for Singapore in the same way as we evaluated the one for Luzern; we first evaluated the performance of the model retrained in Singapore for the event $P_{31-2}$. The model trained for $P_{31-2}$ accurately reproduced the flood map corresponding to the event $P_{46-1}$ (Fig. 6c, Fig. S5, Table S4). Both the spatial distribution of water and the magnitude of the flood depths were correctly simulated, and the performance metrics outperformed the ones for the simulation from the model trained in Zurich: $RMSE_{0.1} = 0.10$ m, $CSI_{0.1, Sgp} = 0.66$ and $CSI_{0.3, Sgp} = 0.67$. Similar to Luzern, the areas with the largest differences between the target and simulated flood map are located either in areas close to water bodies (Fig. 3) or in areas where the model had successfully predicted high water levels (Fig. 6c).

Next, we analyzed the prediction error for three rainfall events in different target water level ranges (Fig. 7b). The error follows a similar pattern as the one from the model for Luzern (Fig. 7a). We found that the model can faithfully reproduce the flood maps in Singapore for new rainfall events as the model produces low errors. It exhibits a slight underprediction of the

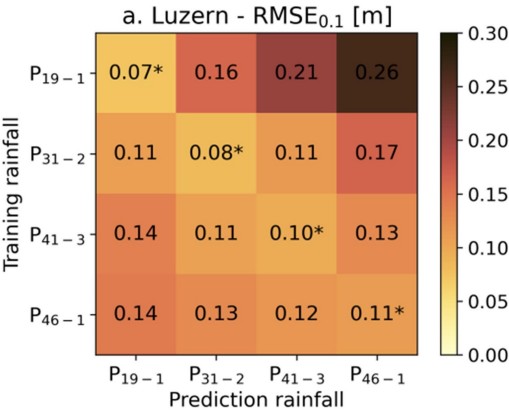 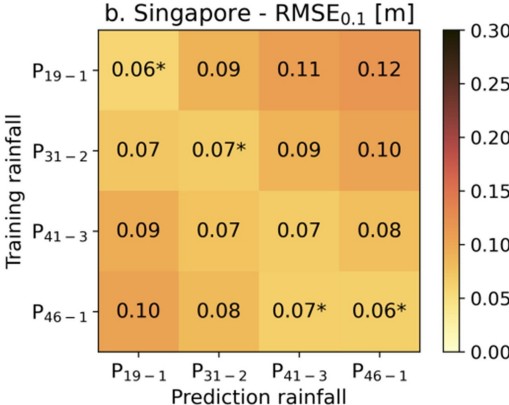

**Figure 8.** Heatmaps of the simulated flood maps $RMSE_{0.1}$ for (a.) Luzern and (b.) Singapore for various rainfall events depending on the rainfall event for which the model was retrained. The asterisks indicate the lowest $RMSE_{0.1}$ for each prediction rainfall.

water levels for the highest range, i.e. for cells with target water depths above 1 m. The error for unseen events, i.e., for $P_{19-1}$

and $P_{46-1}$, is close to the error for $P_{31-2}$ for which the model was trained, which suggests that this model performs well at extrapolating for both less and more intense rainfall events.

Eventually, we compared the $RMSE_{0.1}$ of the simulated flood maps depending on the rainfall event for which the model was trained (Fig. 8b). Similarly as for Luzern, we retrained the model for the rainfall events $P_{19-1}$, $P_{41-3}$, and $P_{46-1}$, and simulated the flood maps for each of these rainfall events and $P_{31-2}$ using all four retrained models. The heatmap of $RMSE_{0.1}$

in Fig. 8b suggests that all retrained models produce accurate results for new rainfall events. Unlike for Luzern, the asterisk is not everywhere along the diagonal of the heatmap. The lowest $RMSE_{0.1}$ for prediction rainfall $P_{41-3}$ was not achieved by the model trained for $P_{41-3}$ but by the model trained for $P_{46-1}$. However, even though this is not consistent with expectations, the $RMSE_{0.1}$ for $P_{41-3}$ simulated by the model trained for $P_{41-3}$ is almost equal to the one achieved by the model trained for $P_{46-1}$. Furthermore, the results suggest that the model can extrapolate equally well to both less and more intense events,

but that the performance of the models decreases as the difference in mean intensity from the training and prediction rainfall events increases.

## 7 Discussion

The proposed context-aware data-driven flood model accurately reproduces high-resolution (1 m) flood maps in the training terrain for unseen extreme rainfall events, with peak rainfall intensities ranging from 42.5mm h$^{-1}$ to 161.4 mm h$^{-1}$. Our model

outperforms other patch-based urban flood emulators (e.g., Guo et al., 2021) mostly in downstream areas and depressions (Table S3), which are critical as these areas will typically be the ones where flooding occurs most.

When simulating flood maps for unseen terrain, the model accurately identifies the locations of water accumulation, which constitutes an improvement compared to the state-of-the-art patch-based model (Fig. S4). This suggests that the model is not overfitting to the training terrain, but extracts the information that captures the hydrologic behaviour of the area. The model adapted well to different spatial resolutions and different representations of the built environment. However, predicting the water levels in unseen terrain remains a challenge. Our results have demonstrated that the model can conveniently adapt to new terrain through the use of transfer learning. After retraining the model for new terrains and for only one rainfall event and its corresponding flood map, the model was effectively adapted to the new terrain while preserving its adaptability to rainfall events. The advantage of this method is that the model can be applied to new terrain without extensive computational resources and training data.

We excluded the water bodies from the simulation results as we want to focus on urban pluvial flooding, and not fluvial flooding for example. However, the model exhibits an acceptable performance level in these areas (Fig. S6), as the model could accurately identify that these areas were flooded.

The data pre-processing framework along with the proposed model architecture has been developed to alleviate the hydrologically counter-intuitive patch-based prediction approach. Yet, this approach remains the most appropriate one for the aim of generalizability to terrain, as sampling (and augmenting) patches increase the number of terrain training images (Romano and Elad, 2016). Consequently, the model can make accurate predictions even in a city with a distinct terrain from the training city, as long as the terrain features in the new city are also present in some areas of the training city. In other words, a model trained on a city that presents a broader variety of topographical features and urban development will probably generalize best. However, we want to emphasize the importance of careful data pre-processing in data-scarce machine learning applications; it is crucial not to over-sample data to avoid overfitting (Fig. S3, Section S1).

We evaluated the model using various combinations of terrain features used in previous studies (Guo et al., 2021; Löwe et al., 2021) such as DEM, mask (a binary image of the catchment area), curvature (plan, profile and mean), aspect (in radians and in degrees), depth of the sinks, slope (in radians and in degrees), flow accumulation (standard, cuberoot transform and weighted by the slope in each cell) and the topographic wetness index (standard and squareroot transform; Löwe et al., 2021). We evaluated only DEM-derived features as other features such as imperviousness or the design of the drainage network might not be freely and easily available. However, if these features influence the hydraulics and hydrology in the training flood maps, their impact will be indirectly captured in the model's predictions. From initial tests (not shown), we found that feeding the model with the DEM, mean curvature, aspect (sine and cosine), depth of the sinks and the slope (in radians) helped the model learn best. Unlike Löwe et al. (2021), we found that using the cuberoot transform of flow accumulation weighted by slope (FLSLO) did not lead to the best performing model. One possible reason is that FLSLO is highly correlated with other terrain variables (e.g. Spearman's rank correlation coefficient of FLSLO with mean curvature and slope are respectively -0.32 and -0.72 in Zurich), while these other terrain variables provide complementary information (Spearman's rank correlation coefficient of mean curvature and slope is 0.02 in Zurich). Additionally, machine learning algorithms often perform poorly on inputs with very different scales (Géron, 2019). This could explain why using non-normalized FLSLO could not improve our

model's performance, while normalizing FLSLO results in the loss of contextual information. Lastly, the terrain features that result in the best performing model may vary depending on the city, and different feature scaling methods could be considered.

Regarding the limitations, machine learning models learn to replicate the errors present in the training data. The model should therefore be trained on the most accurate flood maps available, as the error will propagate from the target flood map onto the simulated flood map. Similar to other studies, the hydrodynamic simulations corresponding to the target flood maps considered 1 h single-peak rainfall events with uniform distribution in space (Guo et al., 2021, 2022), a simplified representation of the drainage network (do Lago et al., 2023; Löwe et al., 2021) and a uniform infiltration rate in space (Guo et al., 2021; Löwe et al., 2021). Additionally, the model was neither trained nor tested for rainfall events with multiple peaks, intermittency, or events with rainfall on more than 1 h. The model should be further tested to account for different types and durations of design storms, reflecting the hyetograph patterns and variability specific to each city. Namely, the model should be evaluated using design storms characteristic of Singapore, such as intense events of up to 2 hours in duration, which align with the island's characteristic heavy rainfall. The lack of representation of the drainage network could represent a limit to the transferability of the model to cities where the drainage network plays a significant role, or in urbanization scenarios where the drainage capacity is changed. On the other hand, the lack of realistic infiltration rates should not be a limitation as urban pluvial floods occur in response to heavy rainfall that becomes saturated and behave as impervious surfaces (Hollis, 1975; Leopold, 1968).

Due to the speed of simulations for standard computing resources (~0.5 s per patch on a 4-core CPU and 16GB of RAM), our model can just as easily be used for flood nowcasting as in the scope of urbanization or climate change impact studies. The model can also be used as a pre-trained model for similar hydrological applications, such as flow velocity mapping (Guo et al., 2022). Some technical improvements to consider are the development of a model that can process different spatial structures of rainfall, as spatial storm profiles can have a significant impact on the flood water depths and areas (Peleg et al., 2022).

Machine learning algorithms have been shown to outperform traditional hydrological models in ungaged catchments, not only in terms of computational time but also in terms of accuracy (Kratzert et al., 2019; Nearing et al., 2021; Zhang et al., 2022). However, this requires a lot of training data to ensure the transfer of information from similar catchments. The lack of an extensive dataset remains a limitation for the development of generalizable models in urban pluvial flooding. Similarly, a benchmark dataset to compare different approaches is still lacking (Bentivoglio et al., 2022). Given the potential of machine learning to address urban flood hazard analysis and early warning, it would be worthwhile for the community to invest efforts in producing and collecting large urban flood databases. These databases could include not only simulated flood data but also observed data from real flood events, made more accessible by recent advancements, such as versatile flood level detection from images (Moy de Vitry et al., 2019), which offer new opportunities for in-situ flood data collection.

## 8 Conclusions

We present a novel context-aware deep learning model for high-resolution urban pluvial flood mapping, which has a 16 times greater visual field than the standard patch-based flood mapping models. The proposed framework is particularly well-suited for flood mapping applications where the continuity of hydrological features, such as flow paths or sinks, is essential. The

model exploits both static (terrain) and dynamic (hyetograph) information to simulate fast urban pluvial flood maps. Our results demonstrate that the model performs well, both in the training terrain (i.e., the same city used for the training) and in new terrains (i.e., application to another unseen city). The context-aware model could simulate accurate results for a variety of rainfall events, with different hyetographs shapes and intensities. When applied to new terrain, the model adapts well to different building representations and spatial resolution. While the generalizability to terrain is not yet fully achieved, we showed that the model accurately identifies the area of water accumulation and that transfer learning is an efficient way to adapt the model to the new terrain.

*Code and data availability.* The source codes, trained models and simulation data are freely available in the following repository: https://doi.org/10.5281/zenodo.10688079 (Cache and Gomez, 2024), and in the GitHub repository https://github.com/tcache1/context_aware_flood_model. We implemented the model and data pre-processing framework in Tensorflow version 2.6.2 (Abadi et al., 2015) using Python version 3.6.13.

*Author contributions.* Conceptualization: TC, NP; software development: TC, MSG; data preparation: TC, JB; formal analyses: TC; funding acquisition: NP; paper writing – original draft: TC; paper writing – review and editing: TC, MSG, TB, JB, JPL, NP.

*Competing interests.* At least one of the (co-)authors is a member of the editorial board of Hydrology and Earth System Sciences.

*Acknowledgements.* TC and NP were supported by the Swiss National Science Foundation (SNSF), Grant 194649 ("Rainfall and floods in future cities"). JB was funded in part by the Future Cities Lab Global programme. Future Cities Lab Global is supported and funded by the National Research Foundation, Prime Minister's Office, Singapore under its Campus for Research Excellence and Technological Enterprise (CREATE) programme and ETH Zurich (ETHZ), with additional contributions from the National University of Singapore (NUS), Nanyang Technological University (NTU), and the Singapore University of Technology and Design (SUTD).

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
