# Peer review of "Enhancing generalizability of data-driven urban flood models by incorporating contextual information"

_Hydrology and Earth System Sciences, 2024_

## Author Response (AR1)

June 24, 2024

Prof. Yue-Ping Xu
Editor
Hydrology and Earth System Sciences
**Ref. No.: HESS-2024-63**

Dear Prof. Yue-Ping Xu,

We would like to thank you for handling our manuscript. The revised manuscript entitled '*Enhancing the generalizability of data-driven urban pluvial flood models*' is enclosed, along with the revised supplementary material and a letter of response to all the remarks made by the reviewers.

All the concerns raised by the reviewers have been addressed point-by-point, providing explanations where necessary and adjusting the manuscript as recommended. We believe that the reviewer's suggestions have allowed us to improve the manuscript, and that the revised manuscript can be considered for publication in *Hydrology and Earth System Sciences*. Listed below are our responses (in blue) to the comments and suggestions of the reviewer. Line numbers refer to the 'track changes' version of the manuscript.

We look forward to your feedback on the revised manuscript.

Sincerely,
Tabea Cache, on behalf of all co-authors

**Reviewer 1:**

The authors propose an improved version of a patch-based CNN that can include the terrain features at multiple scales, to predict the maximum inundation depths for several catchments and rainfall events.

Moreover, they show that the model benefits from transfer learning, even with little data.

The paper is overall well presented and with some interesting and original conclusions. However, there are also several minor changes that I believe would further improve the strengths of the paper and that must be addressed before publication.

We would like to sincerely thank the reviewer for investing their time and effort to evaluate our manuscript, and for their favorable assessment. We highly appreciate the detailed and constructive feedback provided, which was of invaluable support in revising and improving the manuscript.

**General comments:**

In the introduction, there is no mention of other DL models for flood prediction. I would add a longer section that includes the recent developments not only for CNN-based models with terrain generalizability. You could, for example, add some of the following: (Fraehr et al., 2023; Bentivoglio et al., 2023; Berkhahn and Neuweiler, 2024; Liao et al., 2023; Burrichter et al., 2023; He et al., 2023).

We agree with the reviewer that the manuscript lacked the presentation of recent developments in DL models for flood predictions, also for non-CNN-based models. Accordingly, we have added a paragraph to the revised manuscript that includes a discussion on the generalizability potential and limitations of these models. We focused on models that were developed for the same or similar applications as our model, i.e. urban pluvial flood mapping. Additionally, and in response to another comment from Reviewer 1, we have included the paper by do Lago et al. (2023) in our discussion. The paragraph can be found in lines 58-73 of the 'track change' version of the manuscript:

'Löwe et al. (2021) developed a model for urban pluvial flood mapping and evaluated its prediction performance in a city in Denmark at 5 m resolution, reserving approximately 25% of the area for validation and testing. Although these areas were included in the validation, they were excluded from training, and the model still performed well in these areas. In another study, Guo et al. (2022) assessed the terrain generalizability of a data-driven flood model across 656 catchment areas in Switzerland. The model was able to adapt to new catchments, yet it did not incorporate rainfall as an input, limiting its predictions to the single rainfall event used for generating the training flood maps. More recent advances include the work of Seleem et al. (2023), in which a CNN-based approach was compared with a random forest approach for urban pluvial flood mapping in three study areas in Berlin at 1 m spatial resolution. The authors found that the CNN model could benefit from transfer learning to enhance performance in terrain on which the model was not trained. Generalizability to terrain was also investigated by do Lago et al. (2023), who developed a conditional generative adversarial network that distributes a previously known runoff volume over a given catchment. The generator was able to identify cells where the water level exceeds 0.3 m and to predict the water levels for cells below that threshold. Berkhahn et al. (2024) have used autoencoders to compress data contained in flood maps and a recursive time series prediction model to simulate water depth time series in urban areas, at 6 m resolutions. However, they did not consider generalizability to terrain. Lastly, the generalizability of flood models to both rainfall and terrain was investigated by Fraehr et al. (2023) who developed a fast model for flood inundation prediction at 20 m and 1 h resolution and applied it to two study areas with distinct topography'.

In the introduction, you argue that there's a lack of generalizability to unseen case studies but then you also cite several papers that tackle this issue. While I agree that generalizability to unseen case studies is still an area of research, I would not frame it exactly as a gap in the research. This applies also to the generalizability for both terrain and rainfall that you mention in the abstract (lines 3-5) as there are already papers that deal with it, such as do Lago et al. (2023).

We thank the reviewer for pointing out the lack of clarity and agree with the feedback. In addition to the paragraph discussing the recent advances in DL for urban pluvial flood modelling (in lines 58-73), we have clarified the text, both in the abstract and in the introduction of the manuscript. In lines 3-5 of the abstract, we clarified the lack of generalizability as follows: 'Yet the lack of generalizability of urban pluvial flood data-driven models to both varying rainfall and distinctively different terrain, at the fine resolution required for urban flood mapping, still limits their application'. In lines 75-78, a similar clarification was provided: 'Despite the advancements made in recent years in developing data-driven models for flood predictions, there are still some major challenges to overcome. One of them is the generalizability of the models to unseen case studies, including both varying rainfall and distinctively different terrain, at the fine resolution required for urban flood mapping. This limits their application to real-case studies'.

Same goes for the idea of "contextual information". I would argue that this idea only/mainly applies to case studies with higher spatial resolution, as you have in your experiments. Thus, it does not seem fit for a gap in the literature. I would try to reformulate this gap by arguing that for high spatial resolution domains would benefit from including the information at lower spatial resolutions, because there are certain patterns in the topography that cannot be captured with patch-based models that use high-resolution data.

We thank the reviewer for highlighting that the presented contextual information framework mainly applies to case studies that require high spatial resolution. In response, we have revised the text to clarify the necessity of high spatial resolution in the context of flood mapping in urban areas. Specifically, in lines 79 and 90, we emphasized the application to high-resolution flood models by explicitly mentioning that we apply the idea to 'high-resolution' models. Additionally, we included the resolution of the models discussed in the paragraph on recent developments in DL for urban flood mapping. We believe this revision addresses the reviewer's concern and enhances the clarity of our manuscript.

The paper misses all formulas employed in the model. Despite the sufficiently clear Figures 2 and S1, I would recommend adding the key equations employed by your model and also some equations for the metrics that you employ.

Regarding the equations for the metrics, we have added the equation of the loss function used in the model's training and the equations of the performance evaluation metrics (MSE in line 144, RMSE in line 252, and CSI in line 258). Additionally, we have added references to Fig. S1 in the manuscript (see the title of Fig. 2 and line 97) as Fig. S1 includes all operations of the model and the link to the model code (with relevant references and specific equations within) is provided at the end of the manuscript.

All testing metrics are reported via the MSE, which makes the physical interpretation unclear. I would suggest changing the testing metrics to either root mean squared error or mean absolute error, which can both be in meters instead of meters squared.

We agree that using the root mean squared error (RMSE) makes the physical interpretation of the error clearer, and more straightforward. As suggested, we replaced the MSE with the RMSE both in the manuscript and in the supplementary material, and in the text (see lines 251-253, 262-263, 342 and 380) as well as in the figures (Fig. 6, Fig. 8, and Fig. S6).

It's also not clear to me why you consider the MSE only for water depths larger than 0.1 m. This would make more sense for determining a spatial metric, which might be strongly influenced by those small values. But I would keep the regression metric (whichever you end up considering) evaluated over the whole domain, without thresholds.

Indeed, a justification for this choice was missing. This choice was motivated by the combinations of (1) excellent predictions in the cells with target flood levels below 0.1 m (with CSI values for event $P_{19-1}$ and $P_{46-1}$ of respectively 0.98 and 0.97 in Zurich, and as shown in the violinplots of Fig.5); (2) there is a large imbalance in the water levels as shown in the pie charts in Fig. 5 (with target flood map cells below 0.1 m representing respectively 85% and 81% of all cells in Zurich); and (3) the threshold of 0.1 m is often used in urban pluvial flood mapping as it is considered to be the water level at which damage starts to occur (Kaspersen et al., 2017). Similar results apply for Luzern and Singapore as shown in (1) the violinplot in Fig. 7 and (2) the pie charts in Fig. 7. To address this imbalance, we have focused on the evaluation of the RMSE (previously MSE) in the analysis of the results. However, the RMSE for all cells is also provided for Zurich and shows that the RMSE for all cells is between 2 and 3 times lower than the $RMSE_{0.1}$. A justification for this choice was added accordingly in lines 255-263: 'Furthermore, the model accurately identifies cells below 0.1 m (Fig. 5). The critical success index (CSI), which measures the accuracy of the predictions, is defined as the ratio of correctly identified cells (i.e. true positives, TP) to the sum of correctly identified cells, missed target cells (i.e. false negatives, FN), and incorrectly identified cells (i.e. false positives, FP):

CSI = TP/(TP + FN + FP)

The CSI values for $P_{19-1}$ and $P_{46-1}$ are respectively 0.98 and 0.97. Additionally, the majority of the cells in the target flood maps fall below the 0.1 m threshold, representing respectively 84.6% and 80.8% of all cells (Fig. 5). To address this imbalance and evaluate the prediction performance of the model above the critical 0.1 m wet threshold (Seleem et al. 2023, Skougaard et al., 2017), we also evaluated the RMSE values for cells exceeding 0.1 m in the target flood maps. The RMSE values for wet cells are respectively 55.9 x $10^{-3}$ m and 46.8 x $10^{-3}$ m'.

Moreover, you should include some metric to assess the spatial accuracy, for example with the critical success index, which was used in several previous studies (e.g., Löwe et al., 2021; do Lago et al., 2023; Bentivoglio et al., 2023). Please also define all the metrics you employ before you discuss the results.

The assessment of spatial accuracy was measured by the precision score. However, we agree with the reviewer that the use of a metric that is common across various studies is more valuable. We have therefore replaced the precision score with the critical success index and defined the metric in lines 255-258. The replacements can be found in lines 303-304, line 343 and lines 380-381.

I think sections S1 and S2 can be merged within the main text, since they both include some useful information and they are not that long.

We agree with the added value of incorporating some information currently in the supporting material into the manuscript. Specifically, we brought the sections S1 and S2 in the revised version of the manuscript. Section S1 was merged to the manuscript in lines 127-131: 'The output of the RNN is then scaled by the normalized accumulated rainfall. This multiplicative scaling ensures that the rainfall forcing from the RNN is proportional to the accumulated rainfall, resulting in zero forcing when there is zero rainfall. The scaling is defined for rainfall event i as:

$S_i = \frac{P_{i,acc}}{P_{norm}}$, with $P_{i,acc} = \sum_t P_{i,t}$ and $P_{norm} = \sum_t P_{min,t}$ where t is the time step and $P_{min}$ refers to the rainfall event with minimum accumulated rainfall in the training set'.

Section S2 was merged to the manuscript in lines 195-203: 'First, the rainfall events must be partitioned in a way that ensures independence among training, validation, and testing rainfall

events, with respective proportions of 67%, 11% and 22%. Then, the patch characteristics (i.e., patch location and patch augmentation combinations) are randomly divided into training (90%) and validation (10%) sets. Lastly, some of the training data, consisting of the combination of both patch characteristics and rainfall events, are allocated to the validation dataset. Following this workflow, the data in the training and validation sets are allocated in an 80%-20% ratio. This partitioning strategy ensures that the testing rainfall events remain unseen by the model until evaluation, thereby maintaining the validity of the rainfall generalizability assessment. Consequently, the validation set includes the following combinations of data: (1) new rain and new terrain patch, (2) training rain and new terrain patch, and (3) new rain and training terrain patch'.

I also think Table S1 should be included in the main manuscript as it shows a valuable comparison with another model, which is generally lacking in the rest of the paper. On this regard, I believe your paper would benefit by comparing with the study from Guo et al. 2021, which you cite several times. You also state that you outperform this model (line 350), so I would add a longer analysis if you want to claim that, comparing for example different metrics over the whole test dataset, on both models.

We would like to thank the reviewer for their suggestion. However, we would like to keep our manuscript's focus on our model, its applications, and its performance, rather than doing a comparative analysis with other models. Thus, we believe that it is best to keep Table S1 (now Table S3) in the supplementary material. However, we would still like to keep a broad comparison that can convey the essential aspects that demonstrate the improvements compared to a similar single-batch-based model. To support this assertion, we have included a reference to Fig. S4 in line 405. As such, the primary emphasis of our study lies in evaluating the model's generalizability to different terrains while effectively communicating improvement through Table S1 (now Table S3) and Fig. S4 without the need for additional metrics or analyses across the entire test dataset.

**Specific comments:**

Line 6: it is not clear what you refer to with contextual information. This becomes more clear only later on in the paper, but since it's one of your main improvements I would try to clarify it better also in the abstract. This idea of "context" also emerges when you describe your model in section 2. In a similar fashion, I would clarify better what you mean by context. For example, in line 76 "... extract and combine the information from the high-resolution local patch, its context and the rainfall times series to emulate the corresponding flood map" the term context seems very generic. You could use the notion of multi-scale spatial features (that you use in line 79) to help you clarify (or substitute) your notion of context.

We thank the reviewer for pointing this issue out. We agree and revised the text accordingly in line 9 and line 97 by replacing the word 'context' with 'surrounding area'; in line 81 by clarifying 'contextual information from the surrounding terrain' and in line 95 by clarifying 'surrounding terrain (or context)'.

Line 54: I would add in the patch-based methods also the CGAN from do Lago et al. (2023).

We added the reference to this paper in line 55 and have included it in the new paragraph of the introduction discussing the recent developments in DL in urban pluvial flood mapping.

Line 71: I would add at least a reference here.

We added a reference to the paper from Sirinukunwattana et al. (2018) that compares different multi-scale approaches with a single-scale approach for medical image segmentation (in line 89 of the 'track change' manuscript).

End of introduction: not necessarily needed but you could add a paragraph that specifies how the rest of the paper is structured.

We thank the reviewer and appreciate the suggestion, but we prefer not to add this paragraph. Instead, we have split the last paragraph into two paragraphs to underscore the succinct outline of the study.

Line 103: you mention that you include an RNN because it allows you to model rainfall for different events' duration, yet you only consider events of 1 hour. Moreover, depending on the type of RNN you are using (which is not clear from the architecture) you might have a number of outputs which depend on the length of your input sequence, despite it is true that your RNN in theory works with any input length. How would you deal with input hyetographs that have different durations?

We agree with the reviewer that the manuscript was missing an explanation of the feasibility of using the model for rainfall events longer than 1 hour. To test that the RNN could deal with input hyetographs that have different durations, we included some of the rainfall with zero-padding as detailed in lines 234-236 of the manuscript: 'To ensure the model's equivariance to zero-padding and its ability to handle rainfall events of differing durations, we randomly selected three rainfall events with varying mean intensities and introduced random zero-padding at the beginning and end of these events. Lastly, we introduced an event with zero rainfall and a corresponding flood map showing no flooding.'. We revised Fig. 3 accordingly, to showcase the zero-padded events, as well as the event with zero rainfall.

Figure 2: I would add a reference to the more complete version of the architecture that is in the supplementary material, as it helps clarifying how do the scaled dot-product attention and the multi-context fusion work.

We thank the reviewer for this suggestion. The title of Fig. 2 was changed accordingly, to include a reference to Fig. S1: 'A more detailed schematic of the architecture is provided in Fig. S1'.

I would also specify somewhere (and not only in the manuscript) that you don't only take the DEM as input, but also other several ones (despite obtained from the DEM).

The clarification was included accordingly both in the manuscript and in the supplementary material. In line 110 of the manuscript, we clarified: 'The multi-scale image patches, consisting of the terrain features derived from the DEM, are fed to convolutional encoders that are composed of stacked convolutional layers and pooling layers.'; and in the title of Fig. S1 in the supplementary material, we added: 'The 6 channels of the input images are the terrain features derived from the DEM: DEM, mean curvature, aspect (sine and cosine), depth of the sinks and the slope (in radians).'

Line 107: I think that section S1 can be merged with section 2.5, since it is quite small and would help understanding right away what the normalized accumulated rainfall is.

We agree with this suggestion. We found section 2.4 to be a better fit and merged section S1 as follows (lines 127-132): 'The output of the RNN is then scaled by the normalized accumulated rainfall. This multiplicative scaling ensures that the rainfall forcing from the RNN is proportional to the accumulated rainfall, resulting in zero forcing when there is zero rainfall. The scaling is defined for rainfall event i as:

$S_i = \frac{P_{i,acc}}{P_{norm}}$, with $P_{i,acc} = \sum_t P_{i,t}$ and $P_{norm} = \sum_t P_{min,t}$ where t is the time step and $P_{min}$ refers to the rainfall event with minimum accumulated rainfall in the training set. We normalized the scaling in order to avoid vanishing or exploding gradients issues. Ultimately, this procedure led to a model with better performance (Table S2).'

Section S1: it is not clear if Pmin refers to the event with the least accumulated rainfall for all simulations (training and testing) or just the training ones.

This clarification was missing and was added in the text accordingly (lines 130-131): 'rainfall event with minimum accumulated rainfall in the training set'.

Line 108: adding the adjective "locality-aware contextual" to the output of the attention, makes it seem like there is another operation in between. Consider removing it for clarity.

We have rephrased lines 134-135 to: 'Lastly, the scaled RNN output is concatenated with the locality-aware features extracted from the scaled dot-product attention.'. We have kept the adjective 'locality-aware' to provide an explanation of the effect of the scaled-dot-product attention on the features, which we found to be lacking otherwise.

Line 110: please define what you consider as upsampling layer.

We have defined what we consider as an upsampling layer in lines 135-136: 'the decoder consists of stacked upsampling layers, each comprising one deconvolution (convolutional transpose) layer followed by two convolution layers.'

Line 135: I assume that the upscaling is done via a sort of mean pooling, but in the text it's not clear if you are using another strategy.

We thank the reviewer for pointing out that the upscaling method was missing. This was clarified in lines 105-107: 'To reduce computational costs, context images are rescaled to the same size as the local patch (256 x 256) using Lanczos downsampling filters.'

Line 147: I suppose you mean that the model should be equivariant to rotations, i.e., a rotation of the input should result in an equivalent rotation of the output.

We would like to kindly thank the reviewer for noticing this mistake. We replaced the wording accordingly in line 175: 'The model should be equivariant to rotation and flip transformations.

It would be interesting to also see what is the effect of adding this data augmentation to your training dataset (either in the results section or in the supplementary material).

In our study, the effect of adding the data augmentation mainly allows to have the same amount of training data as in similar studies (e.g. 10,000 patches in Guo et al, 2021 and Seleem et al., 2023) while avoiding large patch overlaps. This allows for more independent training and validation patches. Another crucial consideration is related to a scenario where the model would be trained in a city with a predominant slope and flow direction. The data augmentation is important in this case because the model may learn biases that do not accurately reflect flow accumulation otherwise. For example, in a city with a north-south slope, the model might learn that water always accumulates on the lower boundary of the image. We have verified this by training a model with an architecture identical to the one described in the manuscript but adopting a patch sampling procedure similar to the one from previous studies (Guo et al., 2021; Seleem et al., 2023), and testing the model on the city of Zurich, with and without flipping and rotating the city. The procedure and results are detailed in section S1: 'We trained a model with identical architecture to the one described in the manuscript but adopting a patch sampling procedure similar to the one from previous studies (Guo et al., 2021; Seleem et al., 2023). We randomly sampled 10'000 patches without patch overlap threshold, with minimum study area coverage of 10% for each individual patch and without data augmentation, i.e. no flip and rotations of the patches. The training and validation losses of the model trained with this alternative patch sampling procedure are reported in Table S1 and suggest a good performance of the model. However, after further analysis of the model performance, it appears that the model is overfitting. In fact, we evaluated the model under two conditions: (1) using the city of Zurich as

input, (2) using the city of Zurich with some changes in orientation as input (Table S1). For the latter, we tested the model on two orientation changes (randomly selected; Zurich rotated by 90° and Zurich flipped and rotated by 180°) for the rainfall event $P_{46\text{-}1}$. While the city on which the model is tested is identical to the one used to train and validate the model, the orientation of the city is modified. From the metrics reported in Table S1, we see that the model performs well in Zurich when the city's orientation is not modified (MSE = $0.08 \cdot 10^{-3}$ $m^2$ and $RMSE_{0.1}$ = $18.2 \cdot 10^{-3}$ m), but that the performance drops when the input city is flipped and rotated, with an MSE in the order of 100 times the MSE for the city without flips and rotations, and an $RMSE_{0.1}$ in the order of 13 times the $RMSE_{0.1}$ for the city without flips and rotations. This suggests that the model is overfitting to the training patches.'

Line 156: normalization and min-max scaling are not equivalent. Min-max scaling is a type of normalization. Please clarify it in the text.
The text was modified accordingly in line 185: 'Lastly, all multi-channel image patches are transformed through min-max scaling.'

Sections 4 and 5.1 can probably be merged with section 3.
We thank the reviewer for the suggestion. We have carefully considered various structures for the manuscript to ensure a clear flow of the analysis conducted in the study. The current structure appeared to be the most coherent and effective one. We would therefore prefer to keep the structure as it is.

Lines 210-211: the shape notation seems to range from 1 to 9 instead of from 1 to 3, cfr. Fig. 3. In general, the notation $P\_ms$ seems confusing at times.
We kindly thank the reviewer for bringing this typo to our attention. The manuscript has been corrected in line 245: 'Pm-s'. We realize that the missing hyphen in the pre-print could have led to confusion, but we believe that this has now been rectified.

Figure 5: I am not convinced by the overlapping of pie charts and violin plots. I would either separate them in two figures so that the pie chart becomes clearer or improve the legend for the pie chart.
We thank the reviewer for sharing this suggestion. We have enhanced the visual separation between the pie charts and the violinplots by isolating the pie-charts within a rectangle with borders in both Fig. 5 and Fig. 7. Additionally, and following the suggestion of the reviewer, we added the following clarification in the title of the figure: 'The pie charts illustrate the proportion of cells in each water depth range in the target flood maps. The water depth ranges are indicated by different shading levels: lower water depths are represented by more transparent colors, while higher water depths are depicted with darker colors.'

Lines 230-237: I think Table S1 should be included in the main manuscript as it shows a valuable comparison with another model, which is generally lacking in the rest of the paper.
As detailed above, we would like to keep our manuscript's focus on our model, its applications, and its performance, rather than doing a comparative analysis with other models. We would therefore like to keep Table S1 (now Table S3) in the supplementary material to the manuscript.

Line 245: you mention that the resolution for Singapore is 2m. Does this mean that the large-scales are at 4 and 8 meters now? Shouldn't this affect the performance of your deep learning model since you are capturing different processes now?

Yes, the 2 m resolution for Singapore does indeed mean that the large scales are 4 m and 8 m now. This has been clarified in lines 290-291: 'meaning the multi-channel image patches for Singapore have resolutions of 2 m, 4 m and 8 m.'

Additionally, the fact that it did not affect the model's resolution has been emphasized and discussed in lines 295-296 as well as in lines 308-310: 'Despite the differences in terrain, and especially the representation of the built environment in the DEMs and the spatial resolution, the model broadly captures the areas of water accumulation and the flood hazard levels in both new cities.' (lines 295-296) and 'Despite the more pronounced terrain differences between Zurich and Singapore and the different spatial resolutions of the terrain data, with Singapore having a 2 m DEM and both Zurich and Luzern having 1 m DEMs, the CSI values indicate that the model's performance is similar in Luzern and Singapore.' (lines 308-310).

Section 6.1.1: did you also keep the same learning rate? Generally a high learning rate might cause your model's weights to deviate substantially from the pre-trained model.

Indeed, the learning rate was maintained at 0.0001, which is not regarded as a high learning rate. This choice prevents weights from deviating substantially from the pre-trained model. The retraining process was completed in less than 15 epochs for both Luzern and Singapore.

Figure 7: I would specify that this figure is using the transferred model.

We agree with the reviewer that the figure's title was missing the information that the results were related to the model using transfer learning. The title of Fig. 7 was changed as suggested by the reviewer: 'Violinplot of the simulation error in (a.) Luzern and (b.) Singapore, using the models retrained for the event $P_{31-2}$ in each city respectively.'

Lines 367-368: I got a bit confused with the term different topographical features. You could maybe use some different term such as "a broader variety of topographical features".

We kindly thank the reviewer for bringing this lack of clarity in the formulation to our attention. Following the suggestion, we revised the text accordingly in line 421: 'trained on a city that presents a broader variety of topographical features'.

Lines 368-369: do you have any supporting data for this claim? I don't recall seeing in the manuscript any analysis on the amount of overlapping or over-sampling.

The amount of overlapping was demonstrated in Fig. S3 in the supplementary material. We noticed that a reference to this figure was missing in line 423, which has now been added. Furthermore, the text in lines 368-369 of the pre-print to which the reviewer refers are related to the text in lines 180-184 of the 'track change' version of the manuscript. The rationale behind this claim is that if the overlap between the training and validation patches is too important, the two sets are not independent anymore. This has been added in a discussion in section S1, with supporting analysis in Table S1. A reference to section S1 was also added in line 423.

Lines 386-392: I am not sure this analysis adds much to the discussion, though I agree that there is a lack of a common benchmark dataset.

We thank the reviewer for their comment, but we see this paragraph as relevant and important to the manuscript.

Maybe consider also merging discussions and conclusions since there are some overlaps and the conclusions themselves are not too long.

We appreciate the reviewer's suggestion, but we would prefer to maintain the current structure. This allows to have a clear separation between the discussion points and the concluding remarks.

**Technical corrections:**
line 43: there seems to be an extra "("
We could not find the extra '(' to which the reviewer refers.

*References*

Bentivoglio, R., Isufi, E., Jonkman, S. N., and Taormina, R.: Rapid spatio-temporal flood modelling via hydraulics-based graph neural networks, Hydrology and Earth System Sciences, 27, 4227–4246, https://doi.org/10.5194/hess-27-4227-2023, 2023.

Berkhahn, S. and Neuweiler, I.: Data driven real-time prediction of urban floods with spatial and temporal distribution, Journal of Hydrology X, 22, 100 167, 2024.

Burrichter, B., Hofmann, J., Koltermann da Silva, J., Niemann, A., and Quirmbach, M.: A Spatiotemporal Deep Learning Approach for Urban Pluvial Flood Forecasting with Multi-Source Data, Water, 15, 1760, 2023.

do Lago, C. A., Giacomoni, M. H., Bentivoglio, R., Taormina, R., Gomes, M. N., and Mendiondo, E. M.: Generalizing rapid flood predictions to unseen urban catchments with conditional generative adversarial networks, Journal of Hydrology, p. 129276, https://doi.org/https://doi.org/10.1016/j.jhydrol.2023.129276, 2023.

Fraehr, N., Wang, Q. J., Wu, W., and Nathan, R.: Supercharging hydrodynamic inundation models for instant flood insight, Nature Water, 1, 835–843, 2023.

He, J., Zhang, L., Xiao, T., Wang, H., and Luo, H.: Deep learning enables super-resolution hydrodynamic flooding process modeling under spatiotemporally varying rainstorms, Water Research, 239, 120 057, 2023.

Liao, Y., Wang, Z., Chen, X., and Lai, C.: Fast simulation and prediction of urban pluvial floods using a deep convolutional neural network model, Journal of Hydrology, 624, 129 945, https://doi.org/https://doi.org/10.1016/j.jhydrol.2023.129945, 2023.

Löwe, R., Böhm, J., Jensen, D. G., Leandro, J., and Rasmussen, S. H.: U-FLOOD – Topographic deep learning for predicting urban pluvial flood water depth, Journal of Hydrology, 603, 126 898, https://doi.org/https://doi.org/10.1016/j.jhydrol.2021.126898, 2021.

**Reviewer 2:**

This paper proposes a new machine learning architecture for predicting pluvial flood maps with the aim of achieving better transferability of the models between catchments. The authors also explore the value of transfer earning when applying the model to a new city. The paper is generally interesting and the suggested approaches are quite innovative. I do however have several major criticisms that I think should be addressed before publishing.

We would like to thank the reviewer for the thoughtful evaluation of our manuscript and the constructive feedback provided. We have addressed them in detail in the revised version of the manuscript and hereby improved the manuscript.

**Major comments:**

These are:

The paper makes a number of methodological innovations, but their value is not demonstrated anywhere. How do we know if e.g. the combination of three encoders is responsible for the more spatially balanced prediction errors than Guo 2021 or the RNN time series encoder with normalisation? I think some comparisons between old and new model architecture that help us understand what has actually improved + why, should be included in the paper. The transfer learning results could be shortened to make space in the paper.

We kindly thank the reviewer for this comment. We have addressed it through various revisions in the manuscript. First, we have included a comparison of the losses of models with isolated and different methodological innovations in Table S2 of the supplementary material. Additionally, we have detailed the reason for introducing the scaling of the RNN outputs in lines 126-132 and in lines 234-236. This scaling was introduced to prevent non-zero forcing from the RNN when there is zero rainfall. However, as mentioned hereabove, we would like to avoid a lengthy model-to-model comparison and keep our manuscript's focus on our model, its applications, and its performance.

The use case of the approach is not fully clear to me. I can agree with climate impact studies, while I think that the value of urbanisation studies has not been demonstrated. I think the latter would require implementing some changes in the terrain or the runoff behaviour of the same city and testing if the model can still predict the flood maps. This can be addressed by either clarifying the framing in the introduction/discussion, or by including additional results.

We appreciate the reviewer bringing this issue to our attention. We have revised the framing in the discussion of the revised manuscript, as suggested by the reviewer, in lines 447-448: 'The lack of representation of the drainage network could represent a limit to the transferability of the model to cities where the drainage network plays a significant role, or in urbanization scenarios where the drainage capacity is changed'.

Like R1, I also think that some of the supporting information clearly belongs into the paper (Section S2, Take S1)

We agree with the added value of incorporating some information currently in the supporting material into the manuscript. Specifically, we have brought sections S1 and S2 in the revised version of the manuscript (respectively in lines 126-132 and lines 196-203).

**Detailed comments:**

Line 58-66: Another approach to including context information is to engineer features that provide this information on the pixel level. For example providing flow accumulations or similar as input. I'm lacking a sentence on why you think this approach is not good enough and some references

to related work (e.g. Pham 2020, Zhao 2020, there might be newer work that provides better examples)

We kindly thank the reviewer for bringing this gap to our attention. Though using engineered features as input to the model was tested, this had not been outlined in the manuscript. In fact, we have tested the use of flow accumulation, or similar information, as input to the model and we found that this did not lead to the best performing model. We have analyzed FLSLO (flow accumulation weighted by slope; found to lead to best performing model in Löwe et al., 2021) to understand why this was the case and found that one possible reason for this is the high correlation of FLSLO with other terrain variables such as mean curvature and slope (Spearman's rank correlation coefficient of FLSLO with mean curvature and slope are respectively -0.32 and -0.72 in Zurich), while these terrain variables provide complementary information (Spearman's rank correlation coefficient of mean curvature and slope is 0.02 in Zurich).

Another possible reason why FLSLO was not increasing our model's performance is related to the scaling procedure. We tested our model using FLSLO both with city-level scaling and patch-level scaling. The former procedure leads to input patches with very different scales while the latter procedure leads to the loss of information from the surrounding areas, as detailed in lines 187-194: 'We also tested its application across the entire study area, i.e., extracting patches after normalizing the feature images of the full study area, similar to previous studies (Guo et al., 2021; Löwe et al., 2021). While this could help preserve some information about the position of the patch in its larger context, it also forces patches to have values falling in a very small range (e.g., full study area DEM with values between 0 and 1, and DEM patches with values between 0.455 and 0.495), therefore considerably decreasing the performance of the model.')

We agree with the reviewer that a discussion on the topic was missing and included a detailed explanation in the discussion in lines 424-439: 'We evaluated the model using various combinations of terrain features used in previous studies (Guo et al. 2021, Löwe et al. 2021) such as DEM, mask (a binary image of the catchment area), curvature (plan, profile and mean), aspect (in radians and degrees), depth of the sinks, slope (in radians and degrees), flow accumulation (standard, cuberoot transform and weighted by the slope in each cell) and the topographic wetness index (standard and squareroot transform; Löwe et al., 2021). We evaluated only DEM-derived features as other features such as imperviousness or the design of the drainage network might not be freely and easily available. However, if these features influence the hydraulics and hydrology in the training flood maps, their impact will be indirectly captured in the model's predictions. From initial tests (not shown), we found that feeding the model with the DEM, mean curvature, aspect (sine and cosine), depth of the sinks and the slope (in radians) helped the model learn best. Unlike (Löwe et al., 2021), we found that using the cuberoot transform of flow accumulation weighted by slope (FLSLO) did not lead to the best performing model. One possible reason is that FLSLO is highly correlated with other terrain variables (e.g. Spearman's rank correlation coefficient of FLSLO with mean curvature and slope are respectively -0.32 and -0.72 in Zurich), while these other terrain variables provide complementary information (Spearman's rank correlation coefficient of mean curvature and slope is 0.02 in Zurich). Additionally, machine learning algorithms often perform poorly on inputs with very different scales (Géron, 2019). This could explain why using non-normalized FLSLO could not improve our model's performance, while normalizing FLSLO results in the loss of contextual information. Lastly, the terrain features that result in the best performing model may vary depending on the city, and different feature scaling methods could be considered'.

Line 108: Section S1 belongs into the paper. Why is scaling applied to the RNN output (which will be in some latent space) and not the input?

We agree with the reviewer that section S1 fits well into the paper, and we have moved it into the section 2.4 of the manuscript. Additionally, we have included a clarification for the motivation to scale the RNN output; this was motivated by preventing non-zero forcing from the RNN when there

is zero rainfall. These were added as follows in lines 126-130: 'The output of the RNN is then scaled by the normalized accumulated rainfall. This multiplicative scaling ensures that the rainfall forcing from the RNN is proportional to the accumulated rainfall, resulting in zero forcing when there is zero rainfall. The scaling is defined for rainfall event i as:

$S_i = \frac{P_{i,acc}}{P_{norm}}$, with $P_{i,acc} = \sum_t P_{i,t}$ and $P_{norm} = \sum_t P_{min,t}$ where t is the time step and $P_{min}$ refers to the rainfall event with minimum accumulated rainfall in the training set.'

Line 142: the reason for including imperviousness in some of the other models is that it enables us to distinguish that some pixels generate more runoff than others. That is not related to the terrain features that you describe here. In fact, as far as I can see, you are relying on the "glass surface assumption" when calculating runoff. It would be good to clarify this

We kindly thank the reviewer for bringing this point to our attention. We had clarified the use of a uniform infiltration rate in space in line 444 of the manuscript, and justified it as follows in lines 447-450: 'On the other hand, the lack of realistic infiltration rates should not be a limitation as urban pluvial floods occur in response to heavy rainfall that becomes saturated and behave as impervious surfaces (Hollis, 1975; Leopold, 1968)'. Additionally, if these features influence the target flood maps (i.e. if these are accounted for in deriving the training flood maps, which is the case in our study even though the representation is simplified as the imperviousness is uniform), then the model should non-explicitly learn to account for it. We agree with the reviewer that this justification was missing in the manuscript, and we have added it in lines 427-430: 'We evaluated only DEM-derived features as other features such as imperviousness or the design of the drainage network might not be freely and easily available. However, if these features influence the hydraulics and hydrology in the training flood maps, their impact will be indirectly captured in the model's predictions.'

Line 153: Actually, the UFlood paper makes quite a big effort to avoid overlaps between training and validation patches. You can criticise it for using the validation data for both test and validation, but the validation data were fully independent from the training data

We thank the reviewer for noticing this error. We had confused the reference with the one from Seleem et al. (2023) and have implemented the change in the revised manuscript (line 182).

Line 159: This is very interesting. It implies that the convolution kernels must be learning differences between the features, not the absolute values. I think this should be elaborated more in the paper. Do we also see better performance in the results for Zurich? Could we get the same or better results by using localized terrain data (terrain minus minimum elevation in the patch)?

We highly appreciate the reviewer's interest in these findings. To showcase the changes in performance for a model using features scaled on a city scale, we have trained a model with the exact same architecture, data pre-processing and hyperparameters as the ones from the model presented in the manuscript, but with the exception that the features were scaled at city level. The training and validation losses are reported in Table S2, to which a reference was added in line 193 of the 'track change' version of the manuscript.

The results presented in the manuscript were obtained using localized terrain data (i.e. normalization at the patch level), using min-max scaling, which is close to the reviewer's suggested normalization. The type of normalization chosen can be viewed as a hyperparameter of the model, and we expect that it should not change the performance of our model significantly, as long as it is performed on the patch-level. This could be evaluated by retraining multiple models with different normalization techniques, but we prefer not to include this in the paper as we want to avoid a model-to-model comparison.

However, we have revised the manuscript in lines 438-439 to address the reviewer's comment: 'Lastly, the terrain features that result in the best performing model may vary depending on the city, and different feature scaling methods could be considered'.

Line 214: How do you define wet cells? Based on the labels, predictions or both?
We thank the reviewer for bringing this lack of clarity to our attention. Wet cells were defined as cells above the 0.1 m threshold based on the labels (i.e. target flood maps). A clarification was added accordingly in lines 259-260 and line 263: 'in the target flood maps'.

Line 302: Don't we see in the figures that the models even after transfer learning perform quite a bit worse than in Zurich, and therefore we have not yet succeeded in creating models that can be applied in other cities?
We kindly thank the reviewer for their question. The retrained models perform indeed a bit worse in Luzern and Singapore than the model trained and tested in Zurich. However, the absolute median errors for the retrained model in Luzern and Singapore are below 0.1 m for target water depths of up to 1 m (except for event $P_{46-1}$ in Luzern which has a median error of approximately -0.15 m) and an absolute median error are below 0.2 m in cells with target water depths exceeding 1 m (except for event $P_{19-1}$ which has a median error of -0.22 m in Luzern and -0.33 m in Singapore). This is a good performance that shows that the retrained model can be applied in the respective cities, even if the performance is not exactly as good as for the model trained and tested in Zurich.

I'm addition, in Figure 7 I'm missing the results for Luzern and Singapore without transfer learning, so that we can see the impact of the transfer.
We agree with the reviewer of the added value of a figure showing the same results as in Fig. 7 for the model without transfer learning. We have added it in Fig. S5 in the supplementary material and have referred to it in the revised manuscript in line 340 and line 378.

*References*
Pham, B.T., Luu, C., Phong, T. Van, Trinh, P.T., Shirzadi, A., Renoud, S., Asadi, S., Le, H. Van, von Meding, J., Clague, J.J., 2020. Can deep learning algorithms outperform benchmark machine learning algorithms in flood susceptibility modeling? J. Hydrol. 592, 125615. https://doi.org/10.1016/j.jhydrol.2020.125615

Zhao, G., Pang, B., Xu, Z., Peng, D., Zuo, D., 2020. Urban flood susceptibility assessment based on convolutional neural networks. J. Hydrol. 590, 125235. https://doi.org/10.1016/j.jhydrol.2020.125235

---

## Referee Report (RR1)

**Overall**:

The paper presents a timely contribution to the rapid pluvial flood mapping using machine learning. The main novelty lies in the integration of larger "contextual terrain" information with high-resolution local "patches" (context-aware data-driven model), based on concepts from geospatial image segmentation. The methodology is solid and well-researched with multiple case studies (Zurich, Luzern, Singapore), different topographies, DEM resolutions, and rain events with varying return periods (2-100 years). "Transfer learning" from one terrain to another and a parsimonious retrain for new catchments indicates strong generalization potential. The results are well illustrated, using both visual and statistical criteria to evaluate the performance for unseen test data. However, some sections require clarification or additional details to strengthen the paper.

**General/structural comments**:

1. **L46-49**: You describe the need for rapid flood mapping mainly due to the long computational times of hydrodynamic models. To enhance this perspective, consider mentioning other issues that machine learning models can address, such as calibration flexibility and handling uncertainties in large or complex catchments. Also, acknowledge that while advances in computational power can reduce time for hydrodynamic models, they still don't match the speed and parsimony of machine learning for rapid flood mapping.
2. **L63**: Please spell out "CNN" as "Convolutional Neural Network" the first time it is mentioned for readers unfamiliar with the acronym.
3. **Section 2 and Fig. 2**: You reference Fig. 2 only once in Section 2. Some references to details in Fig. 2 in relevant parts of Section 2 would enhance their presentation. Ensure that the main figure and its caption in the text are as complete as those in the supplementary materials.

4. **L125**: Fix the formatting error for the subscript "norm". $S_i = \frac{P_{i,acc}}{P_n orm}$
5. **L203**: Clarify which model you are referring to in "the model to generate flood maps" (is it the proposed model or WCA2D?). Be consistent in using "generation" for target data preparation and "simulation" for the outputs of the data-driven model.
6. **L225**: Clarify that "18 1-hour uniform rainfall" refers to spatially uniform rainfall.
7. **L227**: Specify that "shapes" refers to hyetograph shapes.
8. **Table 1**: This table seems incomplete or needs redesign. Additionally, ensure it is referenced in the text (it currently isn't).

**Figures**:

1. **Fig. 5**: It's difficult to understand the transparency codes without reading the caption. Add a description, such as "(lighter color for shallower depth)" at the end of the text above the pie charts. Also, mention the case study in the caption, or in the title similar to Fig. 7.

2. **Fig. 7**: Apply the same suggestions as for Fig. 5 regarding the transparency. Additionally, use similar y-axis limits for Figs. 5 and 7a-b for easier comparison.
3. **Fig. 8 and Section 6**: Introduce the heatmap (Fig. 8) first and use it to strengthen your justification for choosing the P31-2 rain event for transfer learning.

**Specific clarification**:

1. **L248-252 (CSI Calculation)**: Clarify how "correctly identified cells" are defined, particularly whether this applies only for depths < or > 0.1 m. Given the high CSI values (0.98), it seems no threshold was applied for significant differences between target and simulated depths, but this should be briefly clarified.
2. **RMSE and CSI Values**: Summarize RMSE and CSI values across all case studies and events in a table for transparency and easier comparison.
3. **$RMSE_{0.1}$**: Define this index when first introduced (L333?).
4. **Transfer Learning (Section 6)**: The section title "terrain adaptation" is a bit misleading. Consider renaming it to "Improving generalizability via transfer learning and parsimonious retraining" to better reflect its focus.
5. Additionally, explain your choice of using only one rainfall event (P31-2) earlier in the section, building on Fig. 8 to justify this selection.

**Discussion enhancements**:

1. **Discuss the potential incomparability** across case studies since rainfall ranges differ by location, but seemingly you used same events from Zurich for Singapore. Singapore may receive more intense rainfall over longer period, resulting in higher water depths for larger areas (due to flatter terrain) and potentially higher modeling errors as you suggested for higher water depth ranges. This opens opportunities for further studies on transfer learning for various rainfall lengths and more diverse intensities. If my assumptions are wrong, you could possibly discuss based on right assumptions.
2. **Generalization Discussion**: Consider adding a reflection on combining hydrodynamic models and AI, or training models based on actual flood events and measured water depths. You could also mention the potential of integrating in-situ and remote sensing data to improve model performance when trained against actual observations.
3. **Hybrid AI and Hydrodynamic Models**: As your model uses multiple data-driven methods, a brief discussion on how future work could benefit from hybrid approaches—where machine learning augments traditional hydrodynamic models—would strengthen the paper's relevance for broader applications. If there is a need for such approach, etc.

---

## Author Response (AR2)

September 23, 2024

Prof. Yue-Ping Xu
Editor
Hydrology and Earth System Sciences

**Ref. No.: HESS-2024-63**

Dear Prof. Yue-Ping Xu,

We would like to thank you once again for handling our manuscript. We are pleased that both Referee #1 and Referee #2 have recommended accepting the revised version of the manuscript after we exhaustively addressed their comments in the first round of revisions. Additionally, we appreciate the opportunity to receive further feedback from another reviewer.

The newly revised manuscript entitled '*Enhancing the generalizability of data-driven urban pluvial flood models*' is enclosed, along with a newly revised supplementary material and our response to all new remarks.

We have addressed all the concerns raised by the reviewer point by point, with explanations where needed. The manuscript has been revised following the recommendations while ensuring that the modifications are in line with previous comments from Referee #1 and #2. We believe the revised manuscript is now ready for publication in *Hydrology and Earth System Sciences*. You may find below our responses (in blue) to the comments and suggestions of Referee #3, where line numbers refer to the 'track changes' version of the newly revised manuscript.

We are looking forward to your feedback.

Sincerely,

Tabea Cache, on behalf of all co-authors

**Letter of response**

**Referee #3**

Overall:

The paper presents a timely contribution to the rapid pluvial flood mapping using machine learning. The main novelty lies in the integration of larger "contextual terrain" information with high-resolution local "patches" (context-aware data-driven model), based on concepts from geospatial image segmentation. The methodology is solid and well-researched with multiple case studies (Zurich, Luzern, Singapore), different topographies, DEM resolutions, and rain events with varying return periods (2-100 years). "Transfer learning" from one terrain to another and a parsimonious retrain for new catchments indicates strong generalization potential. The results are well illustrated, using both visual and statistical criteria to evaluate the performance for unseen test data. However, some sections require clarification or additional details to strengthen the paper.

We sincerely thank the reviewer for dedicating their time to evaluate our manuscript and for offering constructive feedback. We greatly appreciate the thoughtful suggestions, which have been addressed in the responses below and were implemented in the newly revised manuscript.

**General/structural comments:**

1. L46-49: You describe the need for rapid flood mapping mainly due to the long computational times of hydrodynamic models. To enhance this perspective, consider mentioning other issues that machine learning models can address, such as calibration flexibility and handling uncertainties in large or complex catchments. Also, acknowledge that while advances in computational power can reduce time for hydrodynamic models, they still don't match the speed and parsimony of machine learning for rapid flood mapping.

We kindly thank the reviewer for this comment. In response, we have added the following remark in lines 46-48 (line number refers to the track-changes version of the manuscript): 'While recent improvements in computational power and more efficient algorithms have reduced the burden of hydrodynamic models, their run times are still insufficient for applications requiring a high number of simulations.' This is linked to the need to address uncertainty, which is further developed in lines 49-51: 'This is problematic, as multiple runs of these models are required per city to account for the large degree of uncertainty in future climate projections and urban development scenarios (Hirsch, 2011; Miller and Hutchins, 2017), necessitating the development of alternative models.'

2. L63: Please spell out "CNN" as "Convolutional Neural Network" the first time it is mentioned for readers unfamiliar with the acronym.

We thank the reviewer for pointing out this issue. We introduced the acronym in line 65, where CNN is mentioned for the first time in the manuscript.

3. Section 2 and Fig. 2: You reference Fig. 2 only once in Section 2. Some references to details in Fig. 2 in relevant parts of Section 2 would enhance their presentation. Ensure that the main figure and its caption in the text are as complete as those in the supplementary materials.

We agree with the reviewer's suggestion and have added multiple references to Fig. 2 in the text (lines 113, 116, 122, and 139). Additionally, some of these references refer to specific sections of the figure, as suggested by the reviewer. These references include: '(see the 'Inputs' and 'Spatial features' panel in Fig. 2)' in line 113 and '(see 'Scaled dot-product attention mechanism' in Fig. 2)' in line 122. We believe that these references will help the reader link the text with the specific sections of the figure.

4. L125: Fix the formatting error for the subscript "norm".

We kindly thank the reviewer for noticing this formatting mistake, which has been corrected (line 134).

5. L203: Clarify which model you are referring to in "the model to generate flood maps" (is it the proposed model or WCA2D?). Be consistent in using "generation" for target data preparation and "simulation" for the outputs of the data-driven model.

The terms 'generation' for the target data and 'simulation' for the output of the machine learning model have been used more consistently in the revised manuscript. To ensure this consistency, changes have been made in the following lines: 17, 212, 219, 301, 346, 350, 478, and 480.

6. L225: Clarify that "18 1-hour uniform rainfall" refers to spatially uniform rainfall.

We agree with the added value of the suggested clarification, which has been added in line 235: '18 1-hour spatially uniform rainfall'.

7. L227: Specify that "shapes" refers to hyetograph shapes.

We thank the reviewer for highlighting that the term 'hyetograph' was missing. The term has been added (line 237).

8. Table 1: This table seems incomplete or needs redesign. Additionally, ensure it is referenced in the text (it currently isn't).

We have redesigned Table 1 and believe it enhances the table's readability, thereby addressing the reviewer's concern. Furthermore, we would like to kindly thank the reviewer for pointing out that the table had not been referenced in the text. The references to Table 1 have been added in lines 294 and 323.

**Figures:**

1. Fig. 5: It's difficult to understand the transparency codes without reading the caption. Add a description, such as "(lighter color for shallower depth)" at the end of the text above the pie charts. Also, mention the case study in the caption, or in the title similar to Fig. 7.

We agree with the reviewer's suggestion and have revised the description of the pie charts accordingly: 'Proportion of cells in each water depth range (lighter colors indicate shallower depths)' in Fig. 5 and Fig. 7. Additionally, we specified the case study in the caption of Fig. 5: 'Violinplot of the simulation error in Zurich […]'.

2. Fig. 7: Apply the same suggestions as for Fig. 5 regarding the transparency. Additionally, use similar y-axis limits for Figs. 5 and 7a-b for easier comparison.

As mentioned in the response hereabove, the description in Fig. 7 was revised to help readers understand the transparency codes. Furthermore, the y-axis limits in Fig. 5 were adjusted to allow for an easier comparison of the results between the case studies, as suggested by the reviewer.

3. Fig. 8 and Section 6: Introduce the heatmap (Fig. 8) first and use it to strengthen your justification for choosing the P31-2 rain event for transfer learning.

We would like to sincerely thank the reviewer for their suggestion. We considered introducing the heatmap earlier in Section 6 (as suggested) but prefer to maintain the current structure, as we believe it provides greater clarity for readers. Additionally, Fig. 8 presents results for models retrained using various rainfall events, not just $P_{31-2}$, which could make the explanation misleading. Moreover, the performances of the models retrained for different rainfall events are overall comparable, and the models retrained for $P_{31-2}$ do not show the best performance. This is

another reason why we believe that introducing the results presented in Fig. 8 earlier on in Section 6 could be misleading.

**Specific clarification:**
1. L248-252 (CSI Calculation): Clarify how "correctly identified cells" are defined, particularly whether this applies only for depths < or > 0.1 m. Given the high CSI values (0.98), it seems no threshold was applied for significant differences between target and simulated depths, but this should be briefly clarified.
We agree with the reviewer that a clarification was needed. The text now reads: 'The CSI for water depths below 0.1 m, i.e. where positive values in Eq. 1 correspond to water depths below 0.1 m, are 0.98 for $P_{19-1}$ and 0.97 for $P_{46-1}$' (lines 264-265). To avoid any confusion with the CSI values for wet cells, we also clarified the text in lines 309-310: 'Considering a wet cell depth threshold of 0.1 m and a flood depth threshold of 0.3 m (i.e. positive values in Eq. 1 correspond to flood depths above 0.1 m and 0.3 m respectively) [...]'.

2. RMSE and CSI Values: Summarize RMSE and CSI values across all case studies and events in a table for transparency and easier comparison.
We have summarized the $RMSE_{0.1}$, $CSI_{0.1}$ and $CSI_{0.3}$ values across all case studies (i.e. in Zurich, Luzern, and Singapore) for the model trained in Zurich as well as for the models retrained in Luzern and Singapore for $P_{31-2}$. The RMSE and CSI values were evaluated for the following rainfall events: $P_{19-1}$, $P_{31-2}$ and $P_{46-1}$. The table has been added to the supplementary material and referenced in the manuscript in lines 314, 353, and 390. Additionally, we have included the $RMSE_{0.1}$ in the figure's titles in Fig. 4, similarly to Fig. 6.

3. RMSE0.1: Define this index when first introduced (L333?).
The meaning of the notation $RMSE_{0.1}$ was indeed missing and is now defined in line 268, where it is first introduced.

4. Transfer Learning (Section 6): The section title "terrain adaptation" is a bit misleading. Consider renaming it to "Improving generalizability via transfer learning and parsimonious retraining" to better reflect its focus.
We thank the reviewer for the suggested section title revision. We have modified the title accordingly to: 'Improving generalizability to terrain through transfer learning and parsimonious retraining' (lines 317-318).

5. Additionally, explain your choice of using only one rainfall event (P31-2) earlier in the section, building on Fig. 8 to justify this selection.
As mentioned in comment n° 3 of the 'Figures' section above, we sincerely thank the reviewer for their suggestion. However, we believe that the current structure makes the text and methodology easier for readers to follow and understand, and therefore prefer to maintain it as it is.

**Discussion enhancements:**
1. Discuss the potential incomparability across case studies since rainfall ranges differ by location, but seemingly you used same events from Zurich for Singapore. Singapore may receive more intense rainfall over longer period, resulting in higher water depths for larger areas (due to flatter terrain) and potentially higher modeling errors as you suggested for higher water depth ranges. This opens opportunities for further studies on transfer learning for various rainfall lengths and more diverse intensities. If my assumptions are wrong, you could possibly discuss based on right assumptions.
We thank the reviewer for pointing out the need to elaborate on this point in the discussion. While the need to further train and test the model for rainfall events with different characteristics (e.g.

multiple peaks and longer durations) was mentioned in the paper, we have developed the discussion and made it more specific to the case studies of the paper. The new discussion elements were incorporated in the previously existing discussion in lines 452-456: 'Additionally, the model was neither trained nor tested for rainfall events with multiple peaks, intermittency, or events with rainfall on more than 1 h. The model should be further tested to account for different types and durations of design storms, reflecting the hyetograph patterns and variability specific to each city. Namely, the model should be evaluated using design storms characteristic of Singapore, such as intense events of up to 2 hours in duration, which align with the island's characteristic heavy rainfall.'

2. Generalization Discussion: Consider adding a reflection on combining hydrodynamic models and AI, or training models based on actual flood events and measured water depths. You could also mention the potential of integrating in-situ and remote sensing data to improve model performance when trained against actual observations.
We agree that this reflection is relevant to our study and have added it in lines 471-473: 'These databases could include not only simulated flood data but also observed data from real flood events, made more accessible by recent advancements, such as versatile flood level detection from images (Moy de Vitry et al., 2019), which offer new opportunities for in-situ flood data collection.' We intentionally kept this reflection brief to ensure alignment with the comments from Reviewer #2 from the previous round of revisions.

3. Hybrid AI and Hydrodynamic Models: As your model uses multiple data-driven methods, a brief discussion on how future work could benefit from hybrid approaches—where machine learning augments traditional hydrodynamic models—would strengthen the paper's relevance for broader applications. If there is a need for such approach, etc.
We kindly thank the reviewer for their valuable suggestion. This is indeed an interesting area of research, and we have therefore revised the text accordingly in lines 72-80, where we believe the discussion fits well: 'Lastly, another promising application of machine learning for rapid flood mapping is the use of hybrid approaches, which combine the advantages of different model types. The advantages of hybrid approaches have been demonstrated in recent studies, including Fraehr et al. (2023), where a fast model was developed by integrating a simplified, physics-based hydrodynamic model, optimized for speed through a coarse computational grid and long computational time steps, with a mathematical model that transforms the flood patterns from the low-fidelity model into those of high-fidelity, non-simplified models. The model's generalizability was tested in two study areas with distinct topographies, using a temporal resolution of 1 h and a spatial resolution of 20 m.'